# AiluRus: A Scalable ViT Framework for Dense Prediction

**Jin Li**[1,†]    **Yaoming Wang**[1,†]    **Xiaopeng Zhang**[2]    **Bowen Shi**[1]
**Dongsheng Jiang**[2]    **Chenglin Li**[1]    **Wenrui Dai**[1]    **Hongkai Xiong**[1]    **Qi Tian**[2*]
[1]Shanghai Jiao Tong University    [2]Huawei Cloud
{deserve_lj, wang_yaoming, sjtu_shibowen, lcl1985,
daiwenrui, xionghongkai}@sjtu.edu.cn;
{zhangxiaopeng12, jiangdongsheng1, tian.qi1}@huawei.com

## Abstract

Vision transformers (ViTs) have emerged as a prevalent architecture for vision tasks owing to their impressive performance. However, when it comes to handling long token sequences, especially in dense prediction tasks that require high-resolution input, the complexity of ViTs increases significantly. Notably, dense prediction tasks, such as semantic segmentation or object detection, emphasize more on the contours or shapes of objects, while the texture inside objects is less informative. Motivated by this observation, we propose to apply adaptive resolution for different regions in the image according to their importance. Specifically, at the intermediate layer of the ViT, we utilize a spatial-aware density-based clustering algorithm to select representative tokens from the token sequence. Once the representative tokens are determined, we proceed to merge other tokens into their closest representative token. Consequently, semantic similar tokens are merged together to form low-resolution regions, while semantic irrelevant tokens are preserved independently as high-resolution regions. This strategy effectively reduces the number of tokens, allowing subsequent layers to handle a reduced token sequence and achieve acceleration. At the output layers, the resolution of the feature map is restored by unfolding the merged tokens for task prediction. As a result, our method significantly accelerates ViTs for dense prediction tasks. We evaluate our proposed method on three different datasets and observe promising performance. For example, the "Segmenter ViT-L" model can be accelerated by 48% FPS without fine-tuning, while maintaining the performance. Additionally, our method can be applied to accelerate fine-tuning as well. Experimental results demonstrate that we can save 52% training time while accelerating 2.46× FPS with only a 0.09% performance drop. The code is available at https://github.com/caddyless/ailurus/tree/main.

## 1   Introduction

Transformers have shown significant advancements in various vision tasks such as image classification [9, 27, 33, 18], object detection [38, 34], and semantic segmentation[26, 4]. Despite their impressive performance across various visual tasks, the complexity of these models poses challenges for fine-tuning and deployment, particularly as their capacity continues to grow [11, 13, 39]. This complexity issue is particularly relevant for dense prediction tasks that require high-resolution input. Efforts have been made to address this challenge by designing efficient ViT models [36, 6, 20, 24, 19, 16, 23, 31, 14, 1, 17]. However, most of these works are primarily focused on classification tasks and are not applicable to dense prediction tasks. Recently, [15] proposed to expedite well-trained ViTs for dense

---

*: Corresponding author. †: Equal contribution. This work was done when Jin Li interned at Huawei Cloud.

prediction tasks through super-pixel clustering. Nevertheless, the clustering operation can be only conducted in the relatively deep layers, resulting in limited acceleration ratio and scalability.

In this paper, we introduce a novel approach, namely **A**daptive reso**lu**tion with spatial-awa**R**e cl**us**tering (**AiluRus**), to accelerate ViTs for dense prediction tasks. We find that dense prediction tasks focus more on the shape or contour of objects rather than the texture. For instance, in segmentation maps, the contour of objects carries crucial information, while the interior regions are filled with the same prediction values, indicating their lower informativeness compared to the boundaries. Motivated by this observation, we propose a token pruning technique that incorporates adaptive resolution to represent different regions in an image. Specifically, we allocate more tokens to critical regions that contribute to decision-making and fewer tokens to less informative regions. The main challenge is to determine a reasonable assignment of resolutions. To address this issue, we utilize density-based clustering algorithms[10, 21] to generate the assignment of each token, where spatial information is incorporated to encourage neighboring tokens to have the same assignment. Tokens that have the same assignments are averaged to produce the representative tokens. These representative tokens could be the original ones, which correspond to informative regions, or the average of several tokens, which correspond to less informative regions. This approach enables us to reduce the length of the token sequence at intermediate layers, thereby accelerating the model. At the output stage, we restore the original resolution for prediction tasks by assigning the value of the representative token to its corresponding regions.

We provide compelling visualizations in Figure Fig. 2 to support the reasonableness of the generated assignments and their limited impact on visual perception. To assess the effectiveness of our proposed method, we adopt the benchmark of [15] and integrate our method into well-trained models without fine-tuning. Our experimental results demonstrate that AiluRus effectively accelerates ViTs and outperforms previous methods, particularly in scenarios with high acceleration ratios. Specifically, AiluRus achieves a 48% increase in FPS for Segmenter ViT-L while maintaining the performance. Moreover, we further apply AiluRus to accelerate the fine-tuning process. Experiments show that AiluRus reduces training time by 52% while achieving a $2.46\times$ increase in FPS with only a 0.09% drop in performance. These findings demonstrate the effectiveness of AiluRus in accelerating ViTs.

In summary, we list our contributions as follows:

- We propose to apply adaptive resolution on the feature map of ViT-based dense prediction tasks for acceleration without fine-tuning.
- We propose to generate the resolution assignments through the proposed spatial-aware DPC algorithm. Visualizations demonstrate that the produced assignments have little influence on visual perception and thus could expedite models without fine-tuning.
- Our proposed AiluRus can be used to accelerate well-trained models or pre-trained models for inference or fine-tuning. Experiments show that AiluRus could significantly accelerate models with a negligible performance drop.

## 2 Related Work

**Vision transformer for dense prediction tasks.** Transformers have gained immense popularity in Natural Language Processing (NLP) tasks, and there have been considerable efforts to extend their success to computer vision tasks. DETR [38] introduced transformers as the detection head in a convolutional neural network, opening up new avenues for utilizing transformers in dense prediction tasks. This work has inspired the development of hybrid-transformer architectures aimed at facilitating dense prediction tasks [3, 30]. Other works have proposed pure transformer architectures, which have achieved significant progress in recent advances [26, 4, 29]. In this paper, instead of proposing a new architecture or framework for dense prediction tasks, we focus on accelerating existing dense prediction methods.

**Efficient vision transformers.** One of the primary strategies for improving the efficiency of Vision Transformers (ViTs) is to reduce the complexity of the self-attention operation. The conventional self-attention operation involves establishing interactions between any two tokens, resulting in quadratic complexity with respect to the number of tokens. To address this challenge, recent approaches aim to approximate the self-attention results through clustering based on the sparse and low-rank properties of self-attention [36, 6, 28].

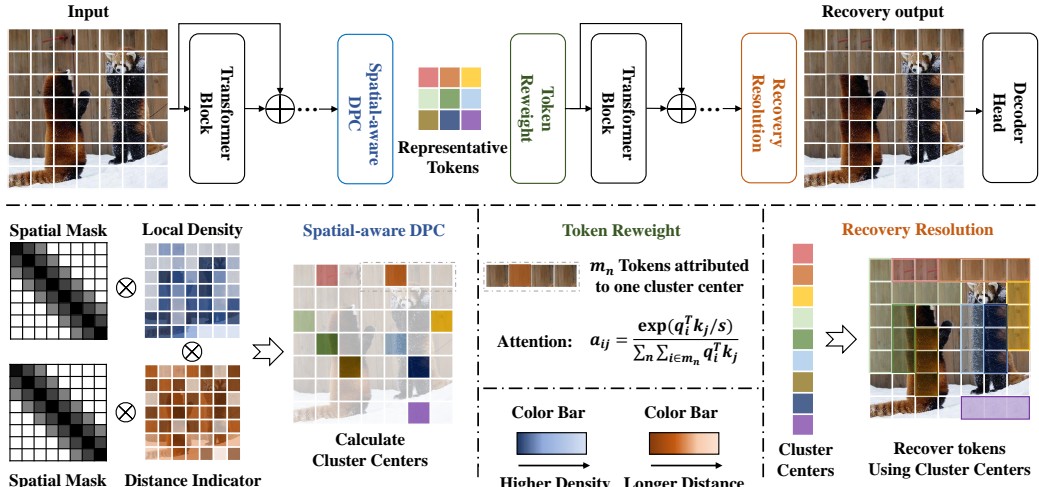

Figure 1: The framework of AiluRus. we focus on a specific intermediate layer of the ViT and apply our AiluRus using the spatial-aware DPC algorithm. This algorithm searches for cluster centers based on local density and distance indicators, and then averages the tokens in each cluster to obtain a representative token for the following layers. To ensure that each representative token is weighted appropriately, we re-weight them based on the number of tokens they represent. Finally, we unfold the representative tokens at the output end to recover the original resolution. For a more detailed explanation of our method, please refer to Section 3.

Another line of works focuses on token pruning [20, 19, 16, 23, 31, 14, 1, 8]. These methods aim to gradually discard less informative tokens at different layers and retain only a subset of tokens at the output end to accelerate the model. However, most of these approaches are designed for classification tasks and are less practical for dense prediction tasks. For instance, EViT [16], Evo-ViT [31], and PCAE [14] select informative tokens based on a single criterion, such as the attention weight to the class token or similarity to the mean token, which is not suitable for dense prediction tasks where there are many objects belonging to various categories in the image. DynamicViT [20] and Ada-ViT [19] rely on specific architectures and require re-training, which may lead to additional computational overhead in practical applications. ToMe [1] progressively merges a percentage of the most similar tokens between bipartite tokens but has only been verified in classification tasks.

Some works aim to design more efficient ViTs by introducing learnable token merging or token pruning modules [2, 12, 32, 25]. For example, STViT [2] replaces redundant tokens with a few semantic tokens, which can be regarded as cluster centers for acceleration. PaCa-ViT [12] proposes patch-to-cluster attention to address the semantic gap between patch-to-patch attention in visual tasks and its NLP counterpart. TCFormer [32] merges tokens from less informative regions via clustering to emphasize critical regions for human-centric tasks. Although these methods achieve promising results for efficient ViTs, some of them rely on specific architectures [32, 12] and all require fine-tuning. Recently, Liang et al. [15] proposed a method to expedite well-trained large ViTs for dense prediction tasks using superpixel clustering, which does not require fine-tuning. However, this strategy can only be applied in relatively deep layers, resulting in limited improvements in efficiency.

## 3 Methodology

### 3.1 Preliminary

A typical ViT requires sequential input and thus reshapes the input image $X \in \mathbb{R}^{H \times W \times 3}$ as a token sequence $X_p \in \mathbb{R}^{(H*W)/p^2 \times 3*p^2}$, where $p$ indicates the patch size. As the patch size is fixed for a given ViT, the number of tokens depends on the resolution of the input image, and thus high-resolution images, which are usually required for dense prediction tasks, inevitably suffer from high computational complexity. One intuitive way toward efficient ViTs is to reduce the number of tokens. However, reducing tokens inevitably accompanies information loss and results in performance degradation. We notice that dense prediction tasks such as detection and segmentation mainly focus on the shape and contour of objects while less caring about the texture inside objects, or irrelevant background. Based on this observation, we propose an adaptive resolution strategy for accelerating dense prediction tasks. Our framework is illustrated in Fig. 1.

## 3.2 Adaptive Resolution

For an input token sequence $Z \in \mathbb{R}^{N \times D}$, the target is to generate $M$ representative tokens according to $Z$ where $M \leq N$. These representative tokens can be either the original tokens in $Z$ that correspond to informative regions in the image or the average of several tokens in $Z$ that correspond to less important regions for decision-making. In this way, different regions are represented by different numbers of tokens, i.e., resolutions based on their importance. The main challenge is to generate a proper assignment of $Z$ that minimizes information loss.

To generate this assignment, we propose to apply density-based clustering algorithms, specifically DPC [21, 10]. We are motivated by two reasons. For one thing, DPC does not rely on iterative updates like traditional clustering methods such as K-means, making it more suitable for latency-sensitive scenarios. For another thing, DPC searches for cluster centers among the input data, and thus specific tokens can be independently preserved, which enables it to preserve details for informative regions. We find that over 20% cluster centers are independently preserved when selecting 400 cluster centers from 1600 tokens (Please refer to supplementary for details). In contrast, the cluster centers in K-means are linearly weighted by input, which will lead to information distortion and affect decision-making for fine-grained regions. However, conventional DPC algorithms only consider the relations of data points in the feature space but ignore their intrinsic spatial structure. Since image patches have clear spatial relationships that are critical for dense prediction, directly applying DPC may lead to unreasonable assignments and performance degradation. To address this issue, we propose to incorporate spatial information into clustering.

**Spatial-aware DPC.** DPC selects cluster centers from input data points based on their product of local density $\rho$ and distance indicator $\delta$, where a large $\rho * \sigma$ indicates the high potential to be the cluster center. We will explain how we calculate $\rho$, $\delta$, and incorporate the spatial information in the following. Specifically, we calculate the local density of each token by:

$$\rho_i = \exp(-\frac{1}{k} \sum_{z_j \in \mathcal{K}} \sigma(z_i, z_j) * s(i, j)) \tag{1}$$

$$s(i, j) = \begin{cases} (1 - \alpha)rank(j)/\lambda + \alpha & rank(j) \leq \lambda \\ inf & rank(j) \geq \lambda \end{cases} \tag{2}$$

where $\sigma(\cdot, \cdot)$ denotes the distance metric, $\mathcal{K} = KNN(z_i)$ and we apply the Euclidean distance here, $k$ indicates the number of neighbors used to calculate the local density, $s(\cdot, \cdot)$ is the introduced spatial information where $\alpha$ is the hyperparameter to control the strength of the spatial constraint, and $\lambda$ is the number of spatial neighbors. $s(\cdot, \cdot)$ assigns different weights for $\sigma(z_i, z_j)$ according to their spatial relation. Specifically, for tokens that are not $\lambda$ nearest, $s(i, j)$ assigns the maximum weight for them while assigning the value of $\alpha$ to 1 for the $\lambda$ nearest tokens. $s(\cdot, \cdot)$ encourages spatially adjacent tokens to be merged first. With the introduced spatial information, the local density of each token only depends on the $\lambda$ spatial neighbors, and each cluster at most merges $\lambda$ tokens. These properties enable the produced assignments to avoid extreme cases where tokens far away in space are merged or too many tokens are merged together.

The distance indicator $\delta$ is calculated by:

$$\delta_i = \begin{cases} \min_{j:\rho_j > \rho_i} \sigma(z_i, z_j) * s(i, j), & \exists \rho_j > \rho_i \\ \inf, & otherwise \end{cases} \tag{3}$$

With $\rho_i$ and $\delta_i$, we rank tokens according to $\rho_i * \delta_i$ and select top $M$ tokens as the cluster centers. The remaining tokens are assigned to the closest cluster centers, and tokens belonging to the same cluster centers are merged together as the representative token.

**Token re-weight.** As the produced representative tokens correspond to different numbers of original tokens (from dozens to only one), there is a distribution gap between the representative tokens and original tokens. For example, tokens corresponding to large objects may be merged into a few representative tokens, which results in inconsistent self-attention results. To minimize this gap, we assign different weights for each representative token during self-attention. In conventional ViTs, the attention of token $i$ to token $j$ is given as:

$$a_{ij} = \frac{\exp(q_i^T k_j / s)}{\sum_i \exp(q_i^T k_j / s)} \tag{4}$$

| Original | Low Resolution | Assignments | Original | Low Resolution | Assignments |
|---|---|---|---|---|---|

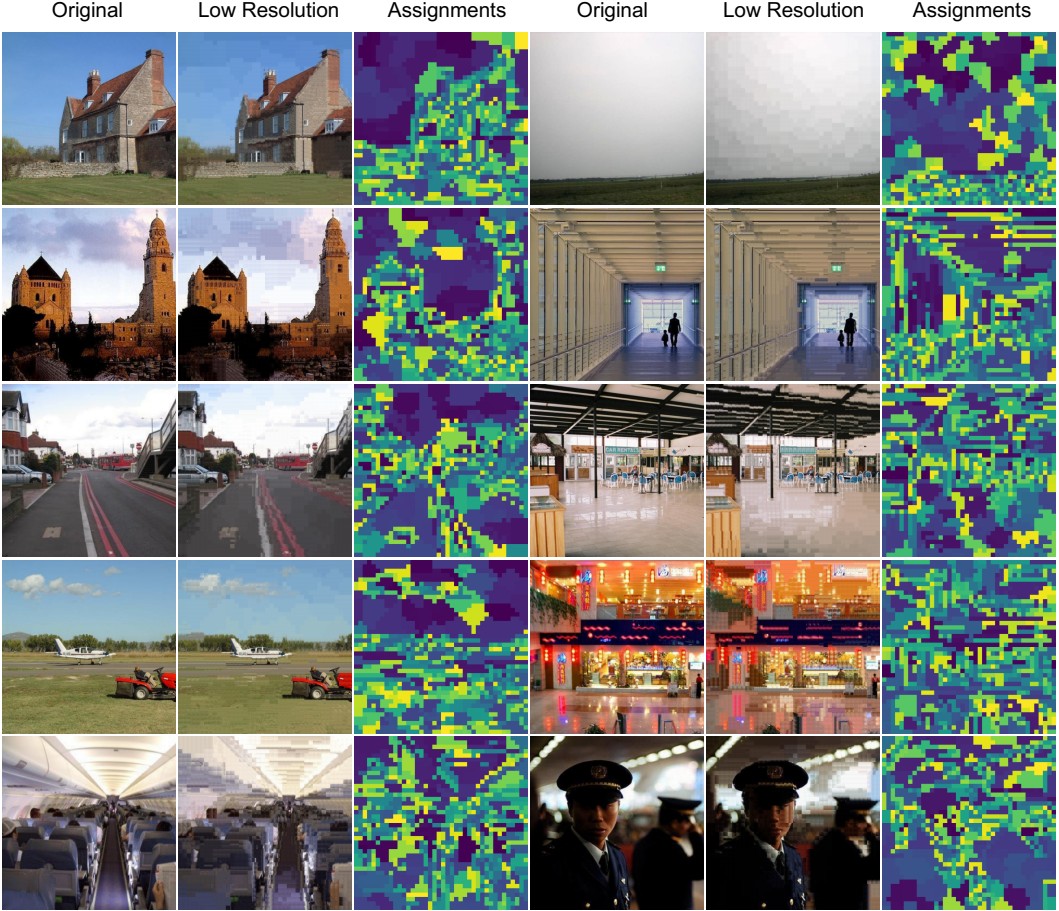

Figure 2: Visualization to clustering results. The first and fourth columns display the original image, the third and sixth columns show the produced assignments, where tokens with the same assignment are marked with the same color, and the reconstructed low-resolution images are presented in the second and fifth columns.

where $q_i$ is the query of token i, $k_j$ is the key of token j and $s$ is the scale constant. To minimize the differences brought by token reduction, the same token is expected to have similar attention values in both the original token sequence and the representative token sequence. To this end, we group tokens belonging to the same representative token in self-attention. As only similar tokens are merged, it can be assumed that their value of $q^T k$ is also similar, and thus Eq. (4) can be written as:

$$a_{ij} = \frac{\exp(q_i^T k_j/s)}{\sum_n \sum_{i \in m_n} q_i^T k_j} \approx \frac{m_n \exp(q_n^T k_j/s)}{\sum_n m_n \exp(q_n^T k_j/s)} \tag{5}$$

where $m_n$ denote the $n_{th}$ representative token. We notice that this trick is also used in [1].

### 3.3 Visualization

To evaluate the effectiveness of the clustering algorithm, we visualize the assignments and the reconstructed low-resolution images. Specifically, we apply our spatial-aware density-based clustering method with 400 cluster centers to the output of the second layer of Segmenter ViT-L, which consists of 1600 tokens. We also reconstruct the entire image using 400 patches corresponding to the cluster centers based on the assignments. The visualizations are shown in Fig. 2. Our results indicate that the reconstructed images have a similar appearance to the original images, suggesting that the produced assignments are reasonable and that 1/4 of the tokens are capable of capturing most of the shape and contour information of the original images. Please note that although some regions may appear to have the same color in the visualization of assignments due to the use of 400 different colors, they may actually have different assignments. These findings provide strong evidence for the effectiveness of our proposed method in generating high-quality representative tokens for dense prediction tasks.

Table 1: GFLOPs, FPS and mIoU of AiluRus under different acceleration ratio. The baseline results refer to Segmenter [26] ViT-L. The best results are marked in bold.

| Methods | Slight | | | Mild | | | Extreme | | |
|---|---|---|---|---|---|---|---|---|---|
| | GFLOPs | FPS | mIoU | GFLOPs | FPS | mIoU | GFLOPs | FPS | mIoU |
| *Results on ADE20k* | | | | | | | | | |
| Baseline | 659.0 | 6.55 | 51.82 | 659.0 | 6.55 | 51.82 | 659.0 | 6.55 | 51.82 |
| ACT [36] | 611.1 | 6.01 | 51.69 | 545.2 | 6.16 | 51.24 | 533.5 | 6.33 | 48.03 |
| ToMe [1] | 516.2 | 6.97 | 51.66 | 448.7 | 8.31 | 50.96 | 321.3 | 10.75 | 47.12 |
| EViT [16] | 572.0 | 7.58 | 51.52 | 500.2 | 8.50 | 50.37 | 351.8 | 12.03 | 38.89 |
| Expedite [15] | 529.8 | 7.92 | 51.93 | 443.8 | 9.51 | 51.56 | 309.4 | 13.51 | 47.96 |
| AiluRus | **478.8** | **8.72** | **52.17** | **427.8** | **9.53** | **51.79** | **300.8** | **14.14** | **50.21** |
| *Results on CityScapes* | | | | | | | | | |
| Baseline | 995.6 | 4.20 | 79.14 | 995.6 | 4.20 | 79.14 | 995.6 | 4.20 | 79.14 |
| ACT [36] | 906.3 | 4.76 | 79.00 | 742.7 | 4.49 | 78.71 | 730.4 | 5.32 | 75.42 |
| ToMe [1] | 760.8 | 5.20 | 78.37 | 651.5 | 5.50 | 77.81 | 448.5 | 7.84 | 71.23 |
| EViT [16] | 822.7 | 5.27 | **79.03** | 707.2 | 5.96 | 78.49 | 506.2 | 8.68 | 68.14 |
| Expedite [15] | 840.9 | 4.82 | 78.82 | 691.0 | 5.89 | 78.38 | 529.6 | 8.02 | 76.20 |
| AiluRus | **710.9** | **5.88** | 78.83 | **669.8** | **6.65** | **78.73** | **461.5** | **9.36** | **77.38** |
| *Results on Pascal Context* | | | | | | | | | |
| Baseline | 338.7 | 14.7 | 58.07 | 338.7 | 14.7 | 58.07 | 338.7 | 14.7 | 58.07 |
| ACT [36] | 306.7 | 11.1 | 58.04 | 299.0 | 11.7 | 57.88 | 298.3 | 11.9 | 56.08 |
| ToMe [1] | 269.8 | 13.9 | 57.67 | 236.5 | 17.0 | 57.24 | 172.4 | 18.2 | 54.25 |
| EViT [16] | 271.7 | 16.0 | 57.94 | 261.0 | 17.7 | 56.99 | 184.4 | 23.5 | 48.57 |
| Expedite [15] | 251.2 | 18.2 | **58.27** | 201.3 | 21.6 | 57.85 | 161.0 | 25.0 | 55.08 |
| AiluRus | **241.2** | **19.8** | 57.95 | **224.3** | **21.9** | **57.91** | **157.7** | **28.4** | **57.02** |

Table 2: GFLOPs, FPS and mIoU of AiluRus under different acceleration ratio. The baseline results refer to Segmenter [26] ViT-B. The best results are marked in bold.

| Methods | Slight | | | Mild | | | Extreme | | |
|---|---|---|---|---|---|---|---|---|---|
| | GFLOPs | FPS | mIoU | GFLOPs | FPS | mIoU | GFLOPs | FPS | mIoU |
| *Results on ADE20k* | | | | | | | | | |
| Baseline | 124.7 | 32.2 | 48.48 | 124.7 | 32.2 | 48.48 | 124.7 | 32.2 | 48.48 |
| ACT [36] | 105.1 | 26.0 | 48.39 | 105.1 | 25.9 | 47.55 | 105.1 | 26.1 | 44.01 |
| ToMe [1] | 100.2 | 31.2 | 47.99 | 88.6 | 32.4 | 46.96 | 66.4 | 35.6 | 40.85 |
| EViT [16] | 109.4 | 34.1 | 48.44 | 96.5 | 37.8 | 48.05 | 69.4 | 46.5 | 38.27 |
| Expedite [15] | 110.2 | 29.8 | 46.74 | 97.4 | 34.6 | 46.14 | 87.1 | 39.4 | 45.17 |
| AiluRus | **62.3** | **37.9** | **48.59** | **50.3** | **45.9** | **48.38** | **40.2** | **55.3** | **47.32** |

## 4 Experiments

In this section, we begin by comparing AiluRus with recent SOTA methods on semantic segmentation tasks in Section 4.1 which includes a more detailed comparison with the reminiscent method [15]. Subsequently, in Section 4.2, we evaluate the performance of AiluRus on object detection and instance segmentation tasks to assess its generalization ability across various dense prediction tasks. Moving on to Section 4.3, we investigate the applicability of AiluRus in the fine-tuning process to enable acceleration. Additionally, in Section 4.4, we conduct ablation experiments to study the impact of different hyper-parameters of AiluRus on the overall performance. Furthermore, we provide supplementary experiments in the appendix, including the application of AiluRus in expediting classification tasks and text-based video generation tasks. We also delve into the reasons behind the superior performance of AiluRus compared to Expedite in these tasks. For more detailed information, please refer to our appendix.

### 4.1 Comparison to other methods

We follow the benchmark in [15] to adapt the proposed method to the Segmenter [26] framework built on ViT [9]. Specifically, we load the parameters of the officially released models and integrate

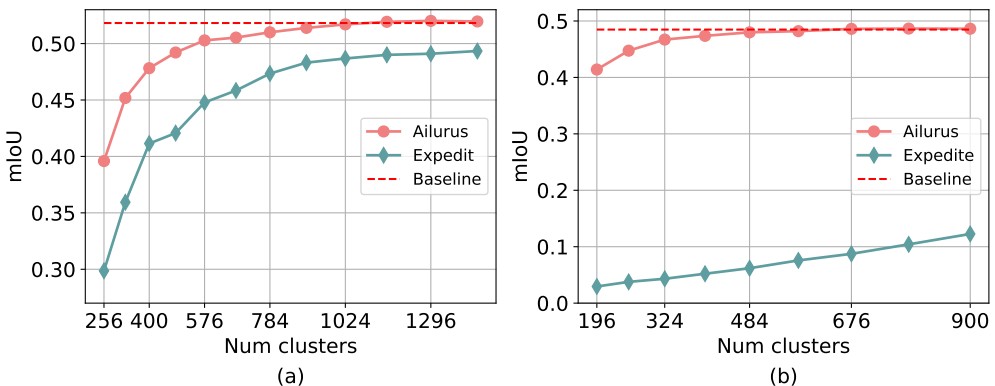

Figure 3: The ablation study on the number of clusters with a fixed cluster location equal to 2. All results are obtained from the officially released checkpoints, and the ablation study on ViT-L and ViT-B are shown in (a) and (b) respectively.

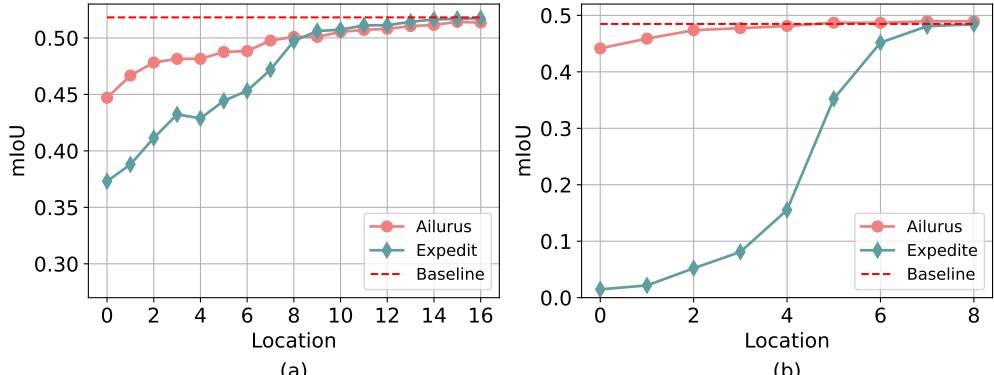

Figure 4: The ablation study on the cluster location with a fixed cluster number equal to 400. All results are obtained from the officially released checkpoints, and the ablation study on ViT-L and ViT-B are shown in (a) and (b) respectively.

AiluRus into the backbone. The adaptive resolution is applied to the output of a certain intermediate layer, and the produced representative tokens are processed by the following layers. With the reduced feature size, the inference speed of models can be significantly improved. We find that the decoder in Segmenter is robust to the reduced feature size, and thus the resolution is recovered after the decoder instead of the backbone.

**Comparisons to SOTA methods.** We conduct experiments to compare AiluRus with recent SOTA efficient ViT methods across different datasets and architectures. The results are presented in Tab. 1 and Tab. 2. Note that Segmenter [26] only provides ViT-B for the ADE20K dataset. Hence, we only compare different methods using ViT-B on the ADE20K dataset. We follow the benchmark set by [15] and report the GFLOPS, FPS, and mIoU under three different acceleration ratios. The results demonstrate that AiluRus consistently achieves the best trade-off between performance and efficiency compared to other methods across various datasets and acceleration ratios. The advantages of AiluRus are particularly evident in the extreme scenario. For instance, as presented in Tab. 1, with higher FPS, AiluRus outperforms Expedite by 2.25↑ mIoU on ADE20K, 1.18↑ mIoU on CityScapes, and 1.94↑ mIoU on Pascal Context, indicating that AiluRus is much more robust to high acceleration ratios. Such a property enables AiluRus achieve more significant acceleration ratios at an acceptable performance drop.

**More comparisons to Expedite.** As both AiluRus and Expedite accelerate ViTs by reducing the feature size at the intermediate layer, they can be further compared under the same configuration. Specifically, we fix either the number of clusters or the cluster location and vary the other parameters across different configurations for both ViT-L and ViT-B models. Initially, we fix the cluster location at 2 and experiment with different numbers of clusters. As illustrated in Fig. 4, AiluRus consistently outperforms Expedite under all settings. The advantage of AiluRus is especially evident for ViT-B,

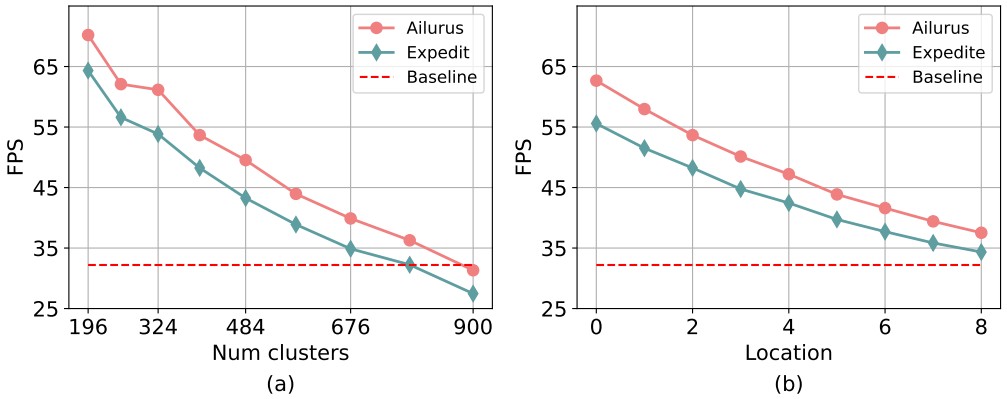

Figure 5: With the officially released 'Segmenter[26] ViT-B', we illustrate the FPS comparison between AiluRus and Expedite under the same configuration.

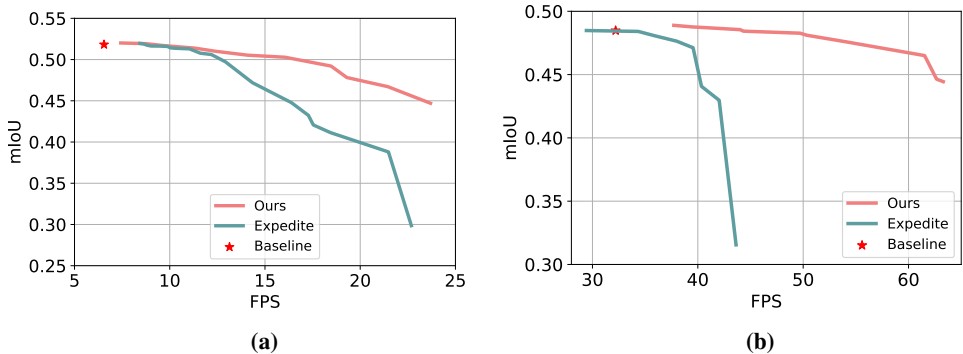

Figure 6: We run different configurations of AiluRus and Expedit (i.e. the cluster number and cluster location) and illustrate the optimal mIoU-FPS curve for ViT-L and ViT-B in figure (a) and (b).

where AiluRus maintains performance even with only 484 clusters, while Expedite almost does not work. Besides the performance, as illustrated in Fig. 5, AiluRus demonstrates higher efficiency than Expedite under the same configuration. This can be attributed to the lower complexity of AiluRus. Unlike Expedite, which requires multiple iterations for accurate cluster centers and inevitably involves high latency, AiluRus selects cluster centers from candidate tokens in one step, making it more suitable for latency-sensitive scenarios. This advantage becomes more pronounced in accelerating relatively lightweight architectures.

Next, we fix the number of clusters at 400 and analyze the impact of cluster location. As depicted in Fig. 3, the performance of AiluRus is close to Expedite when clustering is applied at deeper layers but significantly exceeds Expedite when clustering is performed at shallower layers. This suggests that AiluRus can achieve higher acceleration ratios by reducing the feature map size at shallower layers. In contrast, Expedite suffers serious performance degradation when clustering is applied at shallower layers, limiting its potential in accelerating ViTs.

To make a more comprehensive comparison between AiluRus and Expedite, we run various configurations for AiluRus andExpedite and illustrate the optimal mIoU-FPS curve in Fig. 6. The results consistently demonstrate that AiluRus achieves superior trade-offs between performance and efficiency compared to C, particularly under high acceleration ratios.

## 4.2 Acceleration for object detection and instance segmentation.

To further validate the generalization of AiluRus in dense prediction tasks, we deploy it on object detection and instance segmentation tasks to achieve instant acceleration. Since there are no well-trained models available directly, we follow the setup of CAE and train detection models based on ViT-L and ViT-B within the Mask-RCNN framework. Specifically, we use the CAE pre-trained model as initialization and perform 1x schedule training. The training hyper-parameters strictly

Table 3: The results of AiluRus accelerating Mask-RCNN based on ViT-L.

| Methods | Location | Clusters | FPS | Object Detection | Instance Segmentation |
|---------|----------|----------|-----|------------------|------------------------|
| Baseline | - | - | 1.9 | 54.7 | 47.8 |
| AiluRus | 7 | 1600 | 2.9 (↑53%) | 53.0 (↓1.7) | 46.5 (↓1.3) |
| | 9 | 1764 | 2.6 (↑37%) | 54.0 (↓0.7) | 47.2 (↓0.6) |
| | 11 | 1764 | 2.5 (↑32%) | 54.3 (↓0.4) | 47.4 (↓0.4) |
| | 15 | 1600 | 2.3 (↑21%) | 54.5 (↓0.2) | 47.6 (↓0.2) |

Table 4: The results of AiluRus accelerating Mask-RCNN based on ViT-B.

| Methods | Location | Clusters | FPS | Object Detection | Instance Segmentation |
|---------|----------|----------|-----|------------------|------------------------|
| Baseline | - | - | 4.4 | 50.1 | 44.0 |
| AiluRus | 3 | 1296 | 6.3 (↑43%) | 48.3 (↓1.8) | 42.4 (↓1.6) |
| | 3 | 1764 | 5.9 (↑34%) | 49.4 (↓0.7) | 43.4 (↓0.6) |
| | 6 | 1444 | 5.3 (↑20%) | 49.8 (↓0.3) | 43.9 (↓0.1) |
| | 7 | 1444 | 5.1 (↑16%) | 49.9 (↓0.2) | 43.9 (↓0.1) |

Table 5: The results of AiluRus accelerating fine-tuning based on ViT-L.

| Methods | Location | Clusters | Training Time | FPS | mIoU |
|---------|----------|----------|---------------|-----|------|
| Baseline | - | - | 26.76 | 6.55 | 52.16 |
| Expedite [15] | 2 | 400 | 14.75 (↓45%) | 18.44 (↑182%) | 48.98 (↓3.18) |
| | 2 | 484 | 15.37 (↓43%) | 17.53 (↑168%) | 49.63 (↓2.53) |
| | 2 | 576 | 15.62 (↓42%) | 16.39 (↑150%) | 50.41 (↓1.75) |
| | 2 | 900 | 18.46 (↓31%) | 10.77 (↑64%) | 50.62 (↓1.54) |
| | 8 | 400 | 19.19 (↓28%) | 12.92 (↑97%) | 50.74 (↓1.42) |
| | 10 | 784 | 21.27 (↓21%) | 9.02 (↑38%) | 51.98 (↓0.18) |
| AiluRus | 2 | 400 | 11.80 (↓56%) | 19.24 (↑194%) | 50.82 (↓1.34) |
| | 2 | 484 | 12.28 (↓54%) | 17.30 (↑164%) | 51.82 (↓0.34) |
| | 2 | 576 | 12.81 (↓52%) | 16.11 (↑146%) | 52.07 (↓0.09) |
| | 2 | 900 | 16.37 (↓39%) | 11.24 (↑72%) | 52.42 (↑0.26) |

follow the CAE setup, and the reproduce results are also close to the officially reported ones (reproduce vs official: 50.1 vs 50.3 for ViT-B, 54.7 vs 54.5 for ViT-L). Subsequently, we integrate AiluRus into the well-trained models without fine-tuning, and the hyper-parameters (i.e. $\alpha$, $\lambda$ and $k$) of AiluRus remained the same as in previous experiments. We vary the cluster location and the number of clusters to achieve different acceleration ratios, these results are presented in Tab. 3 and Tab. 4. The results demonstrate that AiluRus generalizes well in object detection and instance segmentation tasks, indicating that AiluRus can effectively expedite various dense prediction tasks with minor modifications.

## 4.3 Acceleration for fine-tuning

The growing capacity of ViTs leads to satisfactory performance but remains a major challenge in fine-tuning the pre-trained models. Since AiluRus does not rely on any specific architecture or learnable parameter, it can be seamlessly integrated into the fine-tuning phase for acceleration. We adopt AiluRus to Segmenter[26] and fine-tune the pre-trained model following the official schedule. Our code base is MMsegmentation [5], and the results reported by MMsegmentation are used as the baseline. We fine-tune the pre-trained modes on 8 V100-32G and evaluate the FPS on single V-100 32G. Both ViT-L and ViT-B are evaluated, and the results are reported in Tab. 5 and Tab. 6. We surprisingly find that AiluRus can largely reduce the overheads in fine-tuning while slightly degrading performance or even enjoying better performance. Specifically, take ViT-L as example, AiluRus reduces 52% training time(↓) and improves 146% FPS(↑) with only 0.09 mIoU drop when remaining 576 clusters. Besides, AiluRus even achieves better performance(↑ 0.26) with improved FPS(↑ 72%) when remaining 900 clusters.

Table 6: The results of AiluRus accelerating fine-tuning based on ViT-B.

| Methods | Location | Clusters | Training Time | FPS | mIoU |
|---|---|---|---|---|---|
| Baseline | - | - | 6.89 | 32.2 | 49.60 |
| Expedite [15] | 2 | 324 | 6.12 (↓11%) | 53.8 (↑67%) | 39.66 (↓9.94) |
| | 2 | 400 | 6.13 (↓11%) | 48.2 (↑50%) | 40.20 (↓9.40) |
| | 2 | 484 | 6.16 (↓11%) | 43.3 (↑34%) | 41.27 (↓8.33) |
| | 2 | 576 | 6.21 (↓10%) | 38.9 (↑21%) | 41.94 (↓7.66) |
| | 6 | 400 | 6.79 (↓1%) | 37.7 (↑17%) | 47.00 (↓2.60) |
| | 7 | 400 | 6.94 (↑1%) | 35.8 (↑11%) | 48.88 (↓0.72) |
| AiluRus | 2 | 324 | 4.94 (↓28%) | 61.5 (↑91%) | 48.06 (↓1.54) |
| | 2 | 400 | 5.14 (↓25%) | 54.1 (↑68%) | 49.04 (↓0.56) |
| | 2 | 484 | 5.33 (↓23%) | 49.7 (↑54%) | 49.39 (↓0.21) |
| | 2 | 576 | 5.48 (↓20%) | 44.0 (↑37%) | 49.57 (↓0.03) |

Table 7: Ablation study of the hyper-parameters.

| Parameter | $\alpha$ | | | | | $\lambda$ | | | | | k | | | |
|---|---|---|---|---|---|---|---|---|---|---|---|---|---|---|
| | 0.6 | 0.7 | 0.8 | 0.9 | 1.0 | 0 | 20 | 30 | 50 | 70 | 1 | 2 | 3 | 4 |
| mIoU | 50.59 | 50.76 | 50.75 | **50.81** | 50.67 | 50.47 | 50.48 | 50.58 | **50.81** | 50.73 | **50.81** | 50.45 | 50.29 | 50.19 |

We also run Expedite under the same configurations for a comprehensive comparison. The results in Tab. 5 and Tab. 6 indicate that Expedite still works worse when reducing feature size at shallower layers even with fine-tuning. We further run Expedite under official configurations and find that its performance is obviously improved. However, the acceleration ratios are also significantly decreased as these configurations reduce feature size at deeper layers. This comparison shows that the advantage of AiluRus over Expedite is more obvious in accelerating fine-tuning. We attribute this to the good robustness of AiluRus for feature size reduction and shallower cluster locations. This property enables AiluRus maintain considerable performance at high acceleration ratios with well-trained models and compensates for the performance drop during fine-tuning. In contrast, we find that the low tolerance to shallow clustering layers of Expedite cannot be addressed by fine-tuning, and ultimately results in limited efficiency improvement.

## 4.4 Ablation Study

We conducted hyper-parameter ablation experiments on the adaptive resolution strategy presented in Section 3.2 using the ADE20K semantic segmentation benchmark and the officially released Segmenter ViT-L/16 [26] checkpoint. For the neighbor weight hyper-parameter $\alpha$, we searched its value from 0.6 to 1.0 (1.0 indicates disabling this hyper-parameter), and the results showed that $\alpha = 0.9$ performed best. Similarly, we searched the value of $\lambda$ from 0 to 70 (0 indicates not using spatial information), and the results showed that $\lambda = 50$ performed best. The ablation results of $k$ indicated that $k = 1$, i.e., choosing the closest token to calculate the local density, performed best.

## 5 Conclusion

The emergence of ViTs has empowered numerous vision tasks but also brought increasing overheads in fine-tuning and inference models. In this paper, we proposed a plug-in strategy, called AiluRus , that can be integrated into well-trained models to immediately accelerate inference without any fine-tuning or to pre-trained models to expedite both fine-tuning and inference. Our experiments demonstrated the advantages of AiluRus over previous methods, particularly in cases with high acceleration ratios. For example, with Segmenter ViT-L [26], AiluRus could accelerate FPS by 45% for the well-trained model with a negligible performance drop (↓ 0.03). For the pre-trained model, AiluRus could reduce fine-tuning time by 52% and accelerate FPS by 146% with a minor performance drop (↓ 0.09). These impressive results lead us to believe that current dense prediction tasks contain significant redundancy that unnecessarily benefits performance and can be removed for significant efficiency improvements. We hope that this consideration could inspire future work in the design of dense prediction tasks.

**Acknowledgement**

This work was supported in part by the National Natural Science Foundation of China under Grant 62250055, Grant 61931023, Grant T2122024, Grant 62125109, Grant 61932022, Grant 61972256, Grant 61971285, Grant 61831018, Grant 62120106007, Grant 62320106003, Grant 62371288, and in part by the Program of Shanghai Science and Technology Innovation Project under Grant 20511100100.

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

# APPENDIX

## A    Statistics of Assignments.

The produced assignments of each token play a crucial role in the trade-off between performance and throughput. Therefore, we present the assignment statistics in Fig. 7a, where we deploy AiluRus on Segmenter ViT-L and perform clustering on the output of the second layer. The produced assignments are collected across the ADE20K [37] validation set. For a given cluster center, the number of tokens belonging to it is denoted by $x$, while $y_1(x)$ and $y_2(x)$ indicate the percentage or the number of cluster centers that dominate $x$ tokens. We run this configuration with varying numbers of cluster centers and illustrate their corresponding $y_1(x)$ and $y_2(x)$ in Fig. 7a.

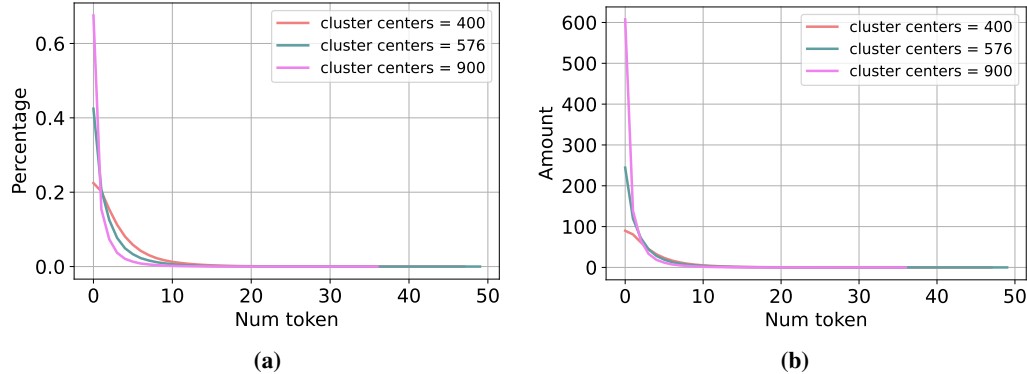

Figure 7:   The statistics of the assignments. The horizontal axis indicates how many tokens each cluster center has. The vertical axis denotes the percentages and amount of a certain kind of cluster centers in (a) and (b) respectively.

The results reveal that cluster centers containing only one token are the most in all cases, and their percentage increases with an increase in the number of cluster centers. As shown in Fig. 7b, the number of cluster centers that dominate more tokens is similar for different total numbers of cluster centers. This suggests that the assignments of the low-resolution regions (dominating more tokens) are similar across different total numbers of cluster centers and the extra cluster centers are assigned to the high-resolution areas. This phenomenon justifies AiluRus because the redundant regions within an image remain fixed, while the areas containing details require more tokens to describe them. Therefore, when increasing the number of cluster centers, the additional cluster centers should be assigned to high-resolution regions to reduce the error.

## B    Acceleration for Classification

Table 8: The results of AiluRus accelerating classification tasks based on DEiT-B [27].

| Methods | Location | Clusters | Throughput (imgs/s) | FLOPs (G) | Accs |
|---|---|---|---|---|---|
| Baseline | - | - | 268.2 | 17.7 | 81.8 |
| Expedite [15] | 2 | 121 | 390.4 (↑46%) | 12.1 (↓32%) | 78.1 (↓3.7) |
| | 4 | 144 | 314.3 (↑17%) | 14.7 (↓17%) | 79.3 (↓2.5) |
| | 6 | 144 | 299.7 (↑12%) | 15.5 (↓12%) | 81.0 (↓0.8) |
| AiluRus | 0 | 121 | 419.8 (↑57%) | 11.4 (↓36%) | 80.4 (↓1.4) |
| | 0 | 144 | 350.5 (↑31%) | 13.3 (↓25%) | 81.3 (↓0.5) |
| | 4 | 144 | 313.7 (↑17%) | 14.9 (↓16%) | 81.7 (↓0.1) |

While we focus on dense prediction tasks in this paper, AiluRus can be easily integrated into classification tasks for instant acceleration. Specifically, we take the typical ImageNet [7] supervised pre-trained DEiT-B as the base model and integrate AiluRus into it without fine-tuning. We run the

same configurations for Expedite [15] in comparison. As presented in Tab. 8, AiluRus consistently outperforms Expedite in various settings.

## C    Acceleration for Video Generation

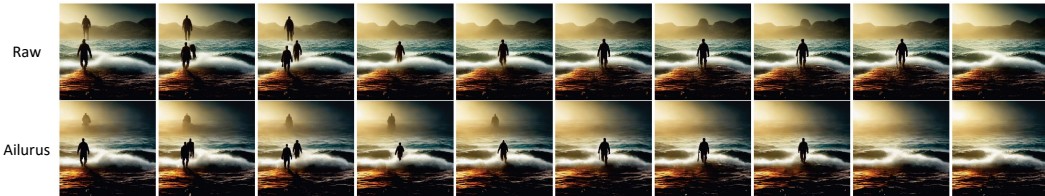

Figure 8: The frames of generated videos. The frames in the first row are generated using the original ControlVideo method [35], while the frames in the second row are generated by integrating AiluRus with ControlVideo.

To demonstrate the versatility of AiluRus, we seamlessly integrate it into the ControlVideo [35] method for text-based video generation. Specifically, we incorporate AiluRus into the StableDiffusion model [22] to reduce the size of intermediate features and execute the model using the prompt "A man wanders on the ocean" with depth condition. We also run the original model with the same configuration for comparison. As shown above, AiluRus produces similar content to the original model. However, AiluRus accomplishes this task in only half the time required by the original model (2:11 vs 4:24). This observation highlights the capability of AiluRus to instantaneously accelerate large models with negligible overheads, which is of critical importance for the recent emergence of large models.

## D    Why AiluRus outperforms Expedite?

As described in the main text, AiluRus exhibits significant performance superiority over Expedite[15], particularly when it comes to token reduction at shallow layers. Shallow layer token reduction poses more significant challenges due to increased information loss. In order to validate this claim, we conduct detailed experiments to explore the intrinsic mechanism.

i) **The reconstruction errors of AiluRus and Expedite**. With the frozen model, the reconstruction errors between the output features and the original features are highly correlated with performance. To provide a comprehensive comparison, we employ various configurations of AiluRus and Expedite on ViT-B and ViT-L models, respectively, and calculate the cosine similarity between the reconstructed features and the original features. The results presented in Tab. 9 and Tab. 10 demonstrate that AiluRus generates output features that are more similar to the original features compared to Expedite across different settings, providing a explanation for the superior performance of AiluRus.

Table 9: The reconstruction similarity comparison between AiluRus and Expedite over ViT-L. X × X indicates the number of clusters.

| Layer | Method | 20×20 | | 22×22 | | 24×24 | | 26×26 | |
|---|---|---|---|---|---|---|---|---|---|
| | | mIoU | Similarity | mIoU | Similarity | mIoU | Similarity | mIoU | Similarity |
| 4 | AiluRus | 47.36 | 0.8401 | 48.84 | 0.8699 | 50.05 | 0.8941 | 50.56 | 0.9143 |
| | Expedite | 42.88 | 0.7706 | 44.26 | 0.7926 | 47.19 | 0.8308 | 47.74 | 0.8504 |
| 6 | AiluRus | 48.80 | 0.8724 | 49.47 | 0.8967 | 50.45 | 0.9164 | 50.98 | 0.9326 |
| | Expedite | 45.32 | 0.8162 | 46.05 | 0.8304 | 47.80 | 0.8626 | 48.37 | 0.8789 |
| 8 | AiluRus | 49.97 | 0.9039 | 50.39 | 0.9213 | 51.06 | 0.9356 | 51.37 | 0.9480 |
| | Expedite | 49.73 | 0.8934 | 50.03 | 0.9048 | 50.77 | 0.9219 | 51.06 | 0.9324 |

ii) **The reconstruction errors for different cluster locations.** We further observe that AiluRus exhibits a more pronounced advantage in shallow clustering, while Expedite's performance deteriorates significantly. Since both AiluRus and Expedite perform clustering only once and reuse the clustering results at the output layer, it is evidently easier to perform clustering in deeper layers compared to

Table 10: The reconstruction similarity comparison between AiluRus and Expedite over ViT-B. X × X indicates the number of clusters.

| Layer | Method | 18×18 | | 20×20 | | 22×22 | | 24×24 | |
|---|---|---|---|---|---|---|---|---|---|
| | | mIoU | Similarity | mIoU | Similarity | mIoU | Similarity | mIoU | Similarity |
| 2 | AiluRus | 46.45 | 0.8971 | 47.12 | 0.9291 | 47.70 | 0.9436 | 48.26 | 0.9602 |
| | Expedite | 4.32 | 0.2882 | 5.21 | 0.3342 | 6.19 | 0.3652 | 7.57 | 0.4065 |
| 4 | AiluRus | 47.69 | 0.9339 | 48.27 | 0.9503 | 48.66 | 0.9636 | 48.88 | 0.9746 |
| | Expedite | 12.66 | 0.4714 | 15.53 | 0.5218 | 18.03 | 0.5594 | 21.57 | 0.6017 |
| 6 | AiluRus | 48.19 | 0.9527 | 48.83 | 0.9642 | 48.90 | 0.9736 | 48.99 | 0.9817 |
| | Expedite | 44.07 | 0.8782 | 45.17 | 0.8972 | 45.44 | 0.9068 | 46.14 | 0.9161 |

Table 11: The reconstruction similarity at different layers.

| Methods | 2 | 3 | 4 | 5 | 6 | 7 | 8 | 9 | 10 | 11 |
|---|---|---|---|---|---|---|---|---|---|---|
| AiluRus | 0.9210 | 0.8959 | 0.8863 | 0.8906 | 0.8972 | 0.9092 | 0.9238 | 0.9312 | 0.9361 | 0.9291 |
| Expedite | 0.7334 | 0.6228 | 0.5524 | 0.4824 | 0.4293 | 0.3694 | 0.3778 | 0.4219 | 0.4433 | 0.3342 |

Table 12: The probability of preserving token pair similarity for different similarity intervals.

| Intervals | 0.0-0.1 | 0.1-0.2 | 0.2-0.3 | 0.3-0.4 | 0.4-0.5 | 0.5-0.6 | 0.6-0.7 | 0.7-0.8 | 0.8-0.9 | 0.9-1.0 |
|---|---|---|---|---|---|---|---|---|---|---|
| Probability | 0.0137 | 0.1246 | 0.2179 | 0.1439 | 0.0629 | 0.0347 | 0.0255 | 0.0193 | 0.0212 | 0.9903 |

shallow layers. To investigate the reasons behind the performance gap in shallow clustering, we analyze a representative scenario (ViT-B, layer index=2, num cluster=20*20) and calculate the feature reconstruction quality in subsequent layers. The results presented in Tab. 11 indicate that Expedite experiences increasingly severe reconstruction distortion during the forward pass, while AiluRus maintains reconstruction quality across layers. This elucidates the poor performance of Expedite when applied in shallow layers.

iii) **Why AiluRus has lower reconstruction errors and maintains its advantage across layers?** We delve into the advantages of AiluRus over Expedite in terms of the intrinsic mechanism. Both methods aim to reduce the number of tokens while preserving the relationships between the retained tokens and the reduced ones, and subsequently utilize the preserved relationships for recovery at the output layer. The quality of recovery determines the overall performance, and it depends on whether the preserved relationships can be maintained after passing through several Transformer blocks. Expedite and AiluRus differ in the nature of the relationship they aim to preserve.

Expedite employs K-means clustering to generate super-pixel features and aims to preserve the relationship between the super-pixel features and original tokens. Since the recovery for each token relies on several super-pixel features, the recorded relationships become complex, and the similarity between the removed token and corresponding super-pixel features is relatively low. In contrast, AiluRus selects representative tokens from the original ones and aggregates the remaining tokens into the nearest representative tokens, which allows AiluRus to recover tokens only relying on the corresponding aggregated token. Therefore, the relationship preserved by AiluRus is simpler, and the similarity between the removed token and the aggregated token is higher. To empirically verify this, we conducted a study on the similarity between the original tokens and the super-pixel features in Expedite, as well as the similarity between the aggregated tokens and the original tokens in AiluRus during the inference process. Specifically, we run Expedite and AiluRus with the same configuration and report the average similarity over the ADE20K validation set. The results confirm that the average similarity in Expedite is 0.6728, while in AiluRus, it is 0.9210, aligning with our expectations.

We further investigate the probability of token pairs in different similarity intervals to maintain their respective intervals at the output end. A higher probability indicates robust preservation of the relationship within that interval, while a lower probability suggests a higher propensity for distortion. Experiments are conducted on the entire validation set of ade20k, and the results are recorded in Tab. 12. The results demonstrate that token pairs with a similarity between 0.9 and 1.0 have a probability of over 99% of maintaining this similarity at the output layer. However, token pairs in other similarity intervals suffer from significant similarity distortion. This finding elucidates why Expedite experiences serious distortion while AiluRus consistently achieves low reconstruction errors across layers.

## E  Latency Analysis

As our motivation is to expedite vision transformers, it is necessary to ensure that the operations introduced by AiluRus are sufficiently efficient, avoiding significant latency that slows down the whole model. To investigate the latency brought by AiluRus , we decompose the running time of models into three components, i.e., time for transformer blocks, time for clustering, and time for recovering, and compare the elapsed time of Expedite and AiluRus. For fair comparisons, we run them under the same configuration over ViT-L and ViT-B respectively. For ViT-L, we start clustering at the 4-th layer and set the number of clusters as 576. For ViT-B, we start clustering at the 2-nd layer and set the number of clusters as 400. We report the average latency of each component on the ADE20K validation set in Tab. 13.

Table 13: The latency analysis. The percentage in parentheses indicates the percentage of latency occupied by the current operation.

| Methods | Total (ms) | blocks (ms) | clustering (ms) | recovering (ms) | mIoU |
|---|---|---|---|---|---|
| *Results on ViT-L* | | | | | |
| Expedite [15] | 31.4 | 26.5 (84.4%) | 3.9 (12.4%) | 1.0 (3.2%) | 47.19 |
| AiluRus | 26.6 | 25.3 (95.1%) | 1.2 (4.5%) | 0.09 (0.3%) | **50.27** |
| *Results on ViT-B* | | | | | |
| Expedite [15] | 11.1 | 8.4 (75.7%) | 2.1 (18.9%) | 0.6 (5.4%) | 5.21 |
| AiluRus | 10.0 | 8.7 (87.0%) | 1.2 (12.0%) | 0.08 (0.8%) | **47.35** |

The analysis results indicate that AiluRus exhibits significantly lower clustering and recovery costs compared to Expedite. As we aforementioned, Expedite replaces raw tokens with super-pixel features for forward passes. Consequently, Expedite relies on multiple iterations to achieve sufficiently accurate super-pixel features and necessitates the computation of reconstruction coefficients for recovering the original features. Thus, Expedite consumes more time in both clustering and reconstruction. In contrast, AiluRus directly operates tokens in the original feature space and thus could efficiently generate token assignments in a single iteration. The produced assignments can be directly used for subsequent reconstruction, thereby avoiding additional recovery costs.

## F  Prediction Visualizations

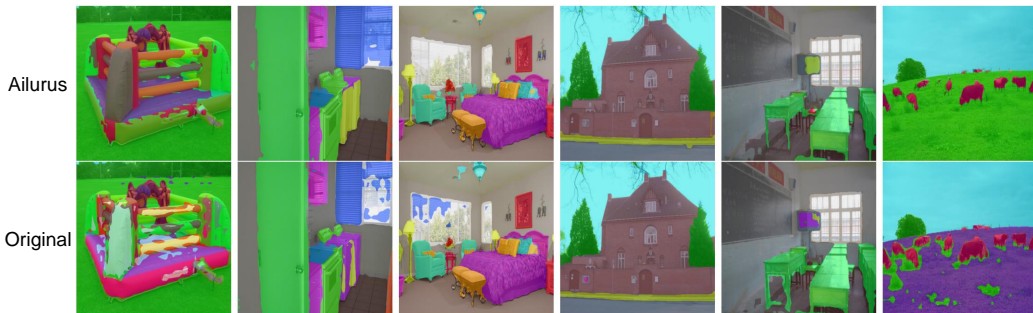

Figure 9:   The prediction visualizations of the original Segmenter ViT-L model and the AiluRus integrated one.

We conducted a comparative analysis of the prediction results between AiluRus and the original model in order to elucidate the factors contributing to the performance improvement of AiluRus in Tab. 5. Specifically, we visualize the predictions generated by both AiluRus and the original models over randomly selected ADE20K [37] images. As illustrated in the figure, the predictions generated by AiluRus exhibit enhanced smoothness, thereby avoiding undesirable predictions such as voids or spikes. Consequently, in certain scenarios, AiluRus demonstrates superiority over the original model. This improved smoothness can be attributed to the application of the adaptive resolution strategy. By introducing a smoothness constraint in regions with low information density, such as backgrounds,

this strategy effectively incorporates a prior for generating smooth predictions in these areas. When retained an adequate number of tokens, this prior aligns well with the true information distribution of the images, leading to performance improvements.

## G   TCFormer for Dense Prediction.

Both TCFormer [32] and AiluRus employ DPC clustering. However, they have very different motivations and implementations. TCFormer is designed for human-centric tasks while AiluRus focuses on expediting dense prediction tasks. In this section, we show that it is untrivial to apply TCFormer for dense prediction tasks.

Table 14:  The latency analysis for TCFormer and AiluRus.

| Method | Total (ms) | blocks (ms) | extra (ms) | mIoU |
|---|---|---|---|---|
| TCFormer | 33.8 | 8.6 (25%) | 25.2 (75%) | 0.14 |
| AiluRus | 10.0 | 8.7 (87%) | 1.3 (13%) | 47.35 |

Table 15:  The performance comparison between AiluRus and ViT-CTM.

| Method | Num clusters | FPS | mIoU |
|---|---|---|---|
| Baseline | 1024 | 32.2 | 49.60 |
| ViT-CTM | 410 (1024 * 0.4) | 29.1 ($\downarrow$ 10%) | 33.48 ($\downarrow$ 16.12) |
| AiluRus | 400 | 53.7 ($\uparrow$ 67%) | 49.04 ($\downarrow$ 0.56) |

Initially, we intend to integrate TCFormer into the mmseg framework for semantic segmentation. We carefully implement the TCFormer under the mmseg framework according to the officially released code. However, we find that TCFormer is an extremely computationally intensive backbone. Even with its lightest configuration, the FPS for inferring ADE20K images at a resolution of 512x512 under the Segmenter framework is only 2.03. In comparison, a typical ViT-Base model achieves an FPS of 32.2, while AiluRus can surpass 50 without sacrificing performance. Upon analyzing the TCFormer code, we discover that it heavily relies on sparse matrix multiplication. These operations may not have high FLOPs yet introducing significant latency. Therefore, it is nearly impossible and impractical to apply TCFormer for dense prediction tasks given its high computational complexity.

Furthermore, we attempt to integrate the core TCFormer design, the CTM module, into ViT for acceleration. This would allow us to compare the performance of TCFormer with AiluRus using the same backbone and initialization. However, we encountered the same issue of excessive complexity for the CTM module. In a similar setup, the time required for the CTM module to execute is even higher than the inference time of the model itself. The details are presented in Tab. 14:

Despite these issues, we proceeded to train the ViT model with the integrated CTM module in the Segmenter framework for comparison with AiluRus. We followed the instructions provided in the TCFormer paper to set up the CTM module and trained ViT-CTM with the same optimization parameters and initialization as AiluRus. However, as demonstrated in Tab. 15, we find that even with only the integration of CTM into ViT, it raises a significant negative impact on training and causes very poor performance. Thus, it is untrivial to integrate the CTM module into ViTs for acceleration.

In conclusion, TCFormer is specifically designed for human-centric tasks, and adapting it to dense prediction tasks poses significant challenges. Additionally, TCFormer exhibits high computational complexity, rendering it unsuitable for accelerating ViTs. In contrast, AiluRus can be seamlessly integrated into well-trained models, providing immediate acceleration or expediting training without the need for additional hyper-parameter adjustments. These two methods differ significantly in terms of motivation, implementation, and application. AiluRus stands out for its flexibility and lightweight nature, enabling its deployment in various tasks that TCFormer is incapable of addressing.

