# Appendix for AiluRus: A Scalable ViT Framework for Dense Prediction

## 1 Acceleration for Object Detection.

We deploy AiluRus to object detection tasks. Specifically, we take the typical Mask-RCNN [2] framework for both object detection and instance segmentation. The pre-trained models of CAE [1] are used for initialization. We run the standard $1\times$ schedule for both ViT-B and ViT-L and the produced results are close to those reported (reproduce vs official: 50.1 vs 50.3 ViT-B, 54.7 vs 54.5 ViT-L). We integrate AiluRus upon the well-trained models without fine-tuning and follow the default hyper-parameter settings. The results are presented in Tab. A-1 and Tab. A-2. The results show that AiluRus still manages to accelerate object detection and instance segmentation tasks with minor performance drops. Additionally, we also find that the instance segmentation task is less sensitive to reduced feature sizes compared to object detection.

## 2 Statistics of Assignments.

The produced assignments of each token play a crucial role in the trade-off between performance and throughput. Therefore, we present the assignment statistics in Fig. A-1a, where we deploy AiluRus on Segmenter ViT-L and perform clustering on the output of the second layer. The produced assignments are collected across the ADE20K [3] validation set. For a given cluster center, the number of tokens belonging to it is denoted by $x$, while $y_1(x)$ and $y_2(x)$ indicate the percentage or the number of cluster centers that dominate $x$ tokens. We run this configuration with varying numbers of cluster centers and illustrate their corresponding $y_1(x)$ and $y_2(x)$ in Fig. A-1a.

The results reveal that cluster centers containing only one token are the most in all cases, and their percentage increases with an increase in the number of cluster centers. As shown in Fig. A-1b, the number of cluster centers that dominate more tokens is similar for different total numbers of cluster centers. This suggests that the assignments of the low-resolution regions (dominating more tokens) are similar across different total numbers of cluster centers and the extra cluster centers are assigned to the high-resolution areas. This phenomenon justifies AiluRus because the redundant regions within an image remain fixed, while the areas containing details require more tokens to describe them. Therefore, when increasing the number of cluster centers, the additional cluster centers should be assigned to high-resolution regions to reduce the error.

## 3 Limitations

Despite its ability to accelerate various dense prediction tasks, AiluRus has some limitations. Our experiments show that AiluRus performs worse in accelerating object detection tasks compared to segmentation tasks. We attribute this to the complex designs used in object detection, such as the FPN and the complicated decoder. AiluRus can be improved by mitigating the performance drop caused by these complex designs. Additionally, AiluRus has only been validated for fine-tuning and inference, and the pre-training phase is taken into account. Integrating AiluRus with existing

Table A-1: The results of AiluRus accelerating Mask-RCNN based on ViT-B.

| Methods | Location | Clusters | FPS | Object Detection | Instance Segmentation |
|---------|----------|----------|-----|------------------|----------------------|
| Baseline | - | - | 4.4 | 50.1 | 44.0 |
| AiluRus | 3 | 1296 | 6.3 (↑43%) | 48.3 (↓1.8) | 42.4 (↓1.6) |
|  | 3 | 1764 | 5.9 (↑34%) | 49.4 (↓0.7) | 43.4 (↓0.6) |
|  | 6 | 1444 | 5.3 (↑20%) | 49.8 (↓0.3) | 43.9 (↓0.1) |
|  | 7 | 1444 | 5.1 (↑16%) | 49.9 (↓0.2) | 43.9 (↓0.1) |

Table A-2: The results of AiluRus accelerating Mask-RCNN based on ViT-L.

| Methods | Location | Clusters | FPS | Object Detection | Instance Segmentation |
|---------|----------|----------|-----|------------------|----------------------|
| Baseline | - | - | 1.9 | 54.7 | 47.8 |
| AiluRus | 7 | 1600 | 2.9 (↑53%) | 53.0 (↓1.7) | 46.5 (↓1.3) |
|  | 9 | 1764 | 2.6 (↑37%) | 54.0 (↓0.7) | 47.2 (↓0.6) |
|  | 11 | 1764 | 2.5 (↑37%) | 54.3 (↓0.4) | 47.4 (↓0.4) |
|  | 15 | 1600 | 2.3 (↑21%) | 54.5 (↓0.2) | 47.6 (↓0.2) |

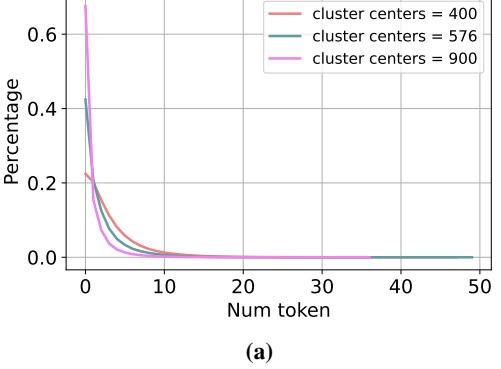 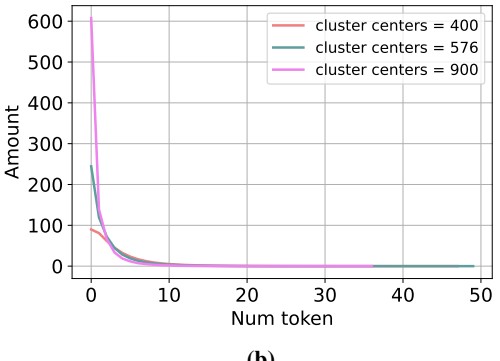

(a)  (b)

Figure A-1: The statistics of the assignments. The horizontal axis indicates how many tokens each cluster center has. The vertical axis denotes the percentages and amount of a certain kind of cluster centers in (a) and (b) respectively.

pre-training methods would benefit it by providing a unified acceleration from large-scale pre-training to practical deployments.

# 4 Reproducibility

We will release our source code once this paper is accepted.