# OpenReview forum: "AiluRus: A Scalable ViT Framework for Dense Prediction"
_NeurIPS.cc/2023/Conference — NeurIPS 2023 poster_

### Official Review · Reviewer_viNG · 2023-06-30

**Soundness:** 3 good
**Presentation:** 3 good
**Contribution:** 2 fair
**Rating:** 4
**Confidence:** 4

**Summary:**

ViTs process input images as a sequence of tokens, and as the token size increases, the computational complexity escalates. The paper aims to accelerate the training and inference of ViTs by reducing the number of tokens. To achieve this, the authors introduce an adaptive resolution strategy, leveraging the observation that dense prediction tasks primarily emphasize object shape and contour. They present a density-based clustering algorithm, spatial-aware DPC, which effectively prunes tokens. The proposed method is extensively evaluated on three diverse datasets, demonstrating its successful acceleration of ViT-based models while preserving performance.

**Strengths:**

- This paper presents an efficient ViT that reduces the complexity of the self-attention operation with a novel dense-based clustering algorithm.
- The paper is well written, and the motivation and technical details are clearly presented.
- The experiments show good qualitative and quantitative results compared to baselines.

**Weaknesses:**

- The idea of merging tokens to speed up ViT is not new (STViT, PaCa-ViT, etc.), and the proposed algorithm relies on DPC, which also reduces the novelty.
- Comparisons are only conducted with Expedite, while there are many other efficient ViTs as introduced in the related works. Though some of them requires fine-tuning, but as least can be compared with AiluRus in table 4.

**Questions:**

- Can the proposed algorithm be applied to classification tasks?
- In Table. 2, the mIoU of AiluRus with 900 clusters outperforms the baseline, I’m interested in the explanation for this.

**Limitations:**

Yes.

---

> ### Author Rebuttal · Authors · 2023-08-10
>
>
> **Q1: The novelty**
>
> **A1:**
> 1. Please note that both STViT and PaCa-ViT rely on training and specific designs, which limits their ability to accelerate well-trained models. In contrast, Ailurus can be seamlessly integrated into pre-trained models for instant acceleration. To illustrate the versatility of Ailurus, we use the recent video generation work, ControlVideo [1], as an example. ControlVideo employs StableDiffusion and ControlNet to generate videos from text. We successfully integrated Ailurus into ControlVideo, reducing the number of tokens during inference by half. The generated results are included in the supplementary PDF file. Notably, Ailurus achieves comparable quality to the original model while significantly reducing the processing time (2:11 vs 4:24). This demonstrates the effectiveness of Ailurus in accelerating large models instantly. Such instant acceleration is unattainable with methodologies like STViT and PaCa-ViT. Moreover, training a new StableDiffusion or other large models using their approaches is nearly impractical due to the architecture gap (Diffusion models mostly use U-nets while STViT and PaCa-ViT are difficult to migrate into U-nets due to their unique designs) and computational overheads. Therefore, it is unreasonable to criticize the novelty of Ailurus based on these methods, as Ailurus offers more promising applications, especially in the current landscape of emerging large models.
>
> 2. As discussed in our main text (Line 37-42) and the general response, Ailurus is motivated by **the heterogeneous distribution of information within images**. Based on this observation, we propose to apply adaptive resolution for the features of different regions. To this end, we need to measure **the information density in different regions** within the image, and DPC, which is based on **local spatial density**, is one of the ways to approach our motivation.
>
> 3. We provide concrete analysis and abundant experiments to show the differences and advantages of Ailurus to previous methods. The results show that Ailurus significantly outperforms previous SOTA methods.
>
> 4. As posted in the general response, DPC is one of the ways to measure information density. Under the framework of Ailurus, it can be replaced with other strategies while keeping satisfactory performance. For example, we replace the DPC clustering with a naive strategy where the information density is measured by the average distance to other tokens. This strategy, termed Ailurus-Dist, still outperforms Expedite significantly as posted below, which demonstrates the effectiveness of Ailurus.
>
> | Method   |   18 * 18 |   20 * 20 |   22 * 22 |   24 * 24 |
> |:---------|----------:|----------:|----------:|----------:|
> | Layer index = 2  |     |    |     |     |
> | Expedite  |    4.32 |    5.21 |    6.19 |    7.57 |
> | Ailurus-Dist  |    36.54  |  40.33 |    43.43 |   45.97 |
> | Layer index = 4  |     |    |     |     |
> | Expedite  |    12.66 |    15.53 |    18.03 |    21.57 |
> | Ailurus-Dist  |    41.89  |  44.14 |    45.79 |   47.29 |
>
> **Q2: Comparison to other efficient ViTs**
>
> **A2：** Please note that we have compared various efficient ViTs in Table 1, and most of them exhibit poor performance on dense prediction tasks. We take Expedite as our main counterpart because it is the SOTA method. Besides, we also demonstrate the versatility of Ailurus by integrating it into ControlVideo for acceleration, which is inaccessible for the mentioned STViT or PaCa-ViT.
>
> **Q3: Classification tasks**
>
> **A3:**
> While we focus on dense prediction tasks in this paper, Ailurus can be easily integrated into classification tasks for instant acceleration. Specifically, we take well-trained DEiT as the base model and integrate Ailurus into it **without finetuning**. We run the same configurations for Expedite
>
> |Method   | Location | Clusters  | Throughput (imgs/s) | FLOPs (G) | Acc |
> |:------| :---------:|:----------:|:--------:|:-----:|:-----:|
> |Baseline   | -   | 196      |    268.2 |  17.7   |81.8   |
> |Expedite     | 6      | 144        |   299.7 ($\uparrow$ 12%) |   15.5 ($\downarrow$ 12%)  | 81.0 ($\downarrow$ 0.8)  |
> |Ailurus       | 4          | 144        |    313.7 ($\uparrow$ 17%) |  14.9 ($\downarrow$ 16%)   | **81.7** ($\downarrow$ 0.1)  |
> |Expedite     | 4      | 144        |  314.3 ($\uparrow$ 17%)   | 14.7 ($\downarrow$ 17%)  | 79.3 ($\downarrow$ 2.5)  |
> |Ailurus       | 0          | 144        |   350.5 ($\uparrow$ 31%) |   13.3 ($\downarrow$ 25%)  | **81.3** ($\downarrow$ 0.5)  |
> |Expedite     | 2      | 121        |   390.4 ($\uparrow$ 46%)     | 12.1 ($\downarrow$ 32%)  |  78.1 ($\downarrow$ 3.7)   |
> |Ailurus       | 0          | 121        |   419.8 ($\uparrow$ 57%) |   11.4 ($\downarrow$ 36%)   | **80.4** ($\downarrow$ 1.4)  |
>
> As posted above, Ailurus still outperforms Expedite on classification tasks.
>
> **Q4: Why Ailurus performs better than baseline?**
>
> **A4:** As stated in our paper, Ailurus employs adaptive resolution for input images. This strategy introduces a smoothness constraint to low information density regions, such as backgrounds, effectively incorporating a prior for smooth predictions in these areas. When retained an adequate number of tokens, this prior aligns well with the true information distribution of the images, resulting in performance gains for the model. We provide visualization of the prediction results of Ailurus and the original one in Figure 1 in supplemented pdf. As depicted in the figure, the predictions generated by Ailurus are more smooth, thereby avoiding terrible predictions such as voids or spikes. Consequently, in certain scenarios, Ailurus demonstrates superiority over the original model.
>
> [1]. Zhang, Yabo, et al. "ControlVideo: Training-free Controllable Text-to-Video Generation." arXiv preprint arXiv:2305.13077 (2023).

---

> > ### Comment · Area_Chair_XNw7 · 2023-08-19
> > **Has the rebuttal addressed your concerns?**
> >
> > Dear Reviewer viNG,
> >
> > Could you please read the author rebuttal and acknowledge if your concerns have been addressed? The discussion period will end very soon on Monday, August 21. Thank you for your time in reviewing this submission!
> >
> > Best,
> >
> > AC

---

### Official Review · Reviewer_2MFv · 2023-07-05

**Soundness:** 3 good
**Presentation:** 3 good
**Contribution:** 3 good
**Rating:** 6
**Confidence:** 3

**Summary:**

This paper first points out that when handling long token sequences, ViTs particularly suffer in terms of computation. Motivated by an observation that dense prediction tasks focus more on the contours or shapes of object, the authors propose to leverage adaptive resolution strategy for different regions depending on each regions' importance. Specifically, regions where repetitive patterns are present, e.g., walls, they are merged to predefined anchors to form low resolution regions, while other "important" regions are preserved for further processing. Thanks to this design, the number of tokens have been greatly reduced, and their shapes are recovered via unfolding operation at the output end. This method has shown to its effectiveness using three different datasets. Especially, for the variant that uses Segmenter ViT-L, almost 48% FPS acceleration is accomplished, which can also benefit to reduce the training time required to learn the models.

**Strengths:**

1. I really like the motivation and idea of this paper. It is a well known issue that the number of tokens fed to ViTs will inevitably require expensive computations and memory footprints, and to me, this paper nicely addresses this issue with a novel approach.

2. This paper is very easy to read and understand.

3. Significant improvements are made when this method is applied.

**Weaknesses:**

1. Figure 2 visualization looks good. I also found the authors saying in L177 that some regions looking like they are assigned to a same group (same color) is not actually in the same group. I believe this visualization in Fig 2. is one of the most important one in this paper, which I would recommend looking for different visualization to make this figure look more clear and easy to catch the idea without reading the texts. Maybe scale the values then visualize? or change the colors?



2. Adding ViT-B results in Table 1 would make this table more informative.

3. What could be the reason for performance improvements in Table 2 last row (AiluRus clusters 900) ?



Minor comments : typo in L200.

**Questions:**

See above weaknesses. I do not have any major issue.

**Limitations:**

"We attribute this to the complex designs used in object detection, such as the FPN and the complicated decoder. AiluRus can be improved by mitigating the performance drop caused by these complex design"

If this was also addressed in this paper, it would have made this paper much stronger. However, this paper already shows good results and analysis in main task (segmentation).

---

> ### Author Rebuttal · Authors · 2023-08-10
>
>
> **Q1: Improve visualization**
>
> **A1:** Thank you for your suggestion! We greatly appreciate it. Based on the concept of coloring a map, we have made improvements to our visualization. Specifically, we use highly distinguishable colors to differentiate adjacent patches belonging to different assignments, while not distinguishing non-adjacent patches. This approach allows us to represent our assignments using a small number of highly distinguishable colors. We have included the updated visualization in the supplementary PDF file. The results demonstrate that our algorithm effectively adheres to our motivation, which is to preserve more tokens in high-information-density regions and fewer tokens in low-information-density regions. This provides a compelling explanation for the outstanding performance of Ailurus. We have incorporated these findings into our revised manuscript.
>
> **Q2: Adding ViT-B results in Table 1**
>
> **A2:** As Segmenter only provide well-trained ViT-B for ADE20K, we add the comparison between different efficient vision transformer methods on ADE20K over ViT-B.
>
> |Method   | GFLOPs | FPS | mIoU | GFLOPs | FPS | mIoU | GFLOPs | FPS | mIoU |
> |:------| :---------:|:----------:|:--------:|:-----:|:-----:|:----------:|:--------:|:-----:|:-----:|
> |Baseline   | 124.7   | 32.2      | 48.48        |124.7   | 32.2      | 48.48         |124.7   | 32.2      | 48.48        |
> |ACT       | 105.1          | 26.0        | 48.39        | 105.1  | 25.9 |    47.55     |    105.1     |  26.1 | 44.01 |
> |ToMe     | 100.2      | 31.2        | 47.99        | 88.58  | 32.4 |   46.96      | 66.4        | 35.6  | 40.85 |
> |EViT       | 109.4          | 34.1        | 48.44        | 96.5  | 37.8 |   48.05      | 69.4        | 46.5  | 38.27 |
> |Expedite     | 110.2      | 29.8        | 46.74        | 97.4  | 34.6 |   46.14      |  87.1        | 39.4  | 45.17 |
> |Ailurus       | **62.3**          | **37.9**        | **48.59**        | **50.3**  | **45.9** |   **48.38**      | **40.2**        | **55.3**  | **47.32** |
>
> As presented above, Ailurus demonstrates even more obvious advantages over previous methods. We owe it to two folds.
>
> First, Ailurus introduces negligible extra overheads. As posted in the response to Reviewer3 PvMg Q2(b), the clustering and recovering operations of Ailurus only take around 1.3 ms while our counterpart, i.e., Expedite takes 2.7 ms which is double our cost. For other methods, ACT and ToMe require to deploy their algorithm at each transformer block, and EViT requires drop tokens three times. As a result, these methods introduce more extra overheads and cause negligible or even negative efficiency improvement at the slight acceleration scenario.
>
> Second, Ailurus is more effective in recovering raw features from reduced ones. As posted in the response to Reviewer gX5G, compared with the recent SOTA method Expedite, Ailurus performs much better in recovering features across various configurations, which empowers Ailurus to accelerate vision transformers with much less performance drop.
>
> Thanks for your suggestion! We have added it to our revised manuscript.
>
> **Q3: The reason for performance improvement**
>
> **A3:** As stated in our paper, Ailurus employs adaptive resolution for input images. This strategy introduces a smoothness constraint to low information density regions, such as backgrounds, effectively incorporating a prior for smooth predictions in these areas. When retained an adequate number of tokens, this prior aligns well with the true information distribution of the images, resulting in performance gains for the model. We provide visualization of the prediction results of Ailurus and the original one in Figure 1 in supplemented pdf. As depicted in the figure, the predictions generated by Ailurus are more smooth, thereby avoiding terrible predictions such as voids or spikes. Consequently, in certain scenarios, Ailurus demonstrates superiority over the original model.

---

> > ### Comment · Area_Chair_XNw7 · 2023-08-19
> > **Has the rebuttal addressed your concerns?**
> >
> > Dear Reviewer 2MFv,
> >
> > Could you please read the author rebuttal and acknowledge if your concerns have been addressed? The discussion period will end very soon on Monday, August 21. Thank you for your time in reviewing this submission!
> >
> > Best,
> >
> > AC

---

> > ### Comment · Reviewer_2MFv · 2023-08-20
> >
> > Thanks for the thorough responses.
> >
> > I've read the rebuttal, and I am still leaning toward acceptance.
> > Therefore, I am keeping my initial rating.

---

### Official Review · Reviewer_PvMg · 2023-07-06

**Soundness:** 2 fair
**Presentation:** 2 fair
**Contribution:** 2 fair
**Rating:** 4
**Confidence:** 4

**Summary:**

This paper proposed a spatial-aware density-based clustering algorithm for ViT-based dense prediction tasks. The proposed method can merge the original tokens into fewer representative tokens to accelerate the transformer inference without finetuning. The experiments on different datasets show that the proposed method can accelerate the ViT inference significantly with a slight performance drop.

**Strengths:**

1. This paper focuses on an important topic of ViT inference acceleration, which may facilitate further applications.
2. The experiments on different datasets prove that the proposed method can speed up both training and inference significantly. The experiments also show that the proposed method can outperform the Expedite method, especially at shallow layers.
3. The visualization results are also reasonable and meaningful.

**Weaknesses:**

1. The technical contribution is incremental compared with the previous method. The DPC-based token clustering module has been proposed in the previous method[29] for transformer-based dense prediction tasks. The only difference between the proposed method in this paper and the previous method is introducing spatial information into the DPC module.

2. The experiments are not sufficient to justify the algorithm design.

    a. The experiment results in Table 3 show that the different settings of α only result in a negligible accuracy variance. The results suggest that the proposed spatial information is incremental to the original DPC.

    b. The DPC module may also introduce computation overhead for each transformer block. It is mandatory to present the break-down time consumption for each block, e.g. the DPC module, the attention module, the feature map restore module, etc.

3. The DPC module will generate representative tokens adaptively for each image. It is not clear whether it will affect the batch inference performance. It is better to show a comparison in the batch inference mode.

4. Some typos and format errors

    a. In L.204, Fig.5 and Fig.4 should be Fig.3 and Fig.4.

    b. In L.217, Fig. 7a and Fig. 7 should be Fig. 7a and Fig. 7b.

    c. The line space between L.154 and L.155 is not sufficient.

**Questions:**

1. Why does the proposed method perform better than existing methods when applied to shallower layers?
2. How does the proposed method recover the output of each transformer block?

**Limitations:**

The authors have adequately addressed the limitations.

---

> ### Author Rebuttal · Authors · 2023-08-10
>
> **Q1: The relation to [29]**
>
> **A1:** We respectfully disagree. TCFormer and Ailurus exhibit significant differences in terms of motivation, implementation, and application scenarios.
>
> 1) The motivation behind TCFormer lies in the utilization of DPC to incorporate saliency information in human-centric scenarios, leading to enhanced performance. Conversely, Ailurus is specifically designed for more general scenarios, with a focus on dense prediction tasks. Ailurus draws motivation from the heterogeneous information distribution within images and applies adaptive resolution to feature maps (please refer to the general response for our detailed motivation), resulting in acceleration for ViTs.
>
> 2) TCFormer relies on specific designs that pose challenges for integration into existing methods. Additionally, the proposed CTM module necessitates training and, therefore, cannot be integrated into well-trained models. Moreover, TCFormer requires the execution of DPC at every layer, rendering it an inefficient approach unsuitable for accelerating ViTs. In contrast, Ailurus can be seamlessly integrated into well-trained models without the need for fine-tuning, enabling instant acceleration.
>
> 3) TCFormer primarily focuses on improving performance in human-centric tasks such as pose estimation, mesh reconstruction, and key point localization. Conventional dense prediction tasks like semantic segmentation have not been adequately explored. In contrast, Ailurus centers its efforts on accelerating well-trained models for dense prediction tasks.
>
> **Q2 (a): The impact of $\alpha$.**
>
> **A2 (a):** As described in the context, the variable $\alpha$ represents the penalty strength applied to tokens within the neighborhood. A smaller value of $\alpha$ indicates a larger penalty strength, indicating a preference for prioritizing closer tokens. Since the tokens are already constrained within a neighborhood, they inherently exhibit relatively high similarity, resulting in relatively low variance for different values of $\alpha$.
>
> **Q2 (b): The break-down time consumption**
>
> **A2 (b):** We decompose the running time into three components, i.e., time for transformer blocks, time for clustering, and time for recovering, and compare the elapsed time of Expedite and Ailurus.
> For fair comparisons, we run Expedite and Ailurus under the same configuration over ViT-L. For ViT-L, we start clustering at 4-th layer and set the number of clusters as 576.
> The results are presented below:
>
> Results for ViT-L
> |Method   | Total (ms) | blocks (ms) | clustering (ms)| recovering (ms)  | mIoU |
> |:------| :---------:|:----------:|:--------:|:-----:|:-----:|
> |Expedite   | 31.4   |   26.5      | 3.9        | 1.0  | 47.19 |
> |Ailurus     | 26.6      | 25.3        | 1.2        | 0.09  | 50.27 |
>
> The above results indicate that Ailurus exhibits significantly lower clustering and recovery costs compared to Expedite. As we analyzed in the general response, Expedite replaces raw tokens with super-pixel features for forward passes. Consequently, Expedite relies on multiple iterations to achieve sufficiently accurate super-pixel features and necessitates the computation of reconstruction coefficients for recovering the original features. Thus, Expedite consumes more time in both clustering and reconstruction. In contrast, Ailurus directly operates tokens in the original feature space and thus could efficiently generate token assignments in a single iteration. The produced assignments can be directly used for subsequent reconstruction, thereby avoiding additional recovery costs.
>
> **Q3: The batch inference performance of Ailurus**
>
> **A3:** It is important to note that our approach employs adaptive resolution for different regions within an image, rather than adapting the number of tokens for different images. Consequently, each image is converted into an equal number of tokens, enabling efficient batch processing. In order to showcase the potential of Ailurus for batch inference, we conducted an investigation to determine the maximum throughput of both Ailurus and the original model on a single V-100 GPU by progressively increasing the batch size. The results revealed that Ailurus achieves its maximum throughput with a batch size of 32, whereas the original model attains its maximum throughput with a batch size of 16. The detailed findings are presented below:
>
> |Method   | Num clusters | Batch size |  FPS | Throughput  | mIoU |
> |:------| :---------:|:----------:|:--------:|:-----:|:-----:|
> |Baseline   | 1024   | 16 |   32.2      | 35.6        | 48.48  |
> |Ailurus     | 324          | 32 | 61.2 ($\uparrow$ 90%)       | 81.6 ($\uparrow$ 129%)       | 46.45  |
> |     | 400      | 32 | 53.7 ($\uparrow$ 67%)       | 69.6 ($\uparrow$ 96%)        | 47.12  |
>
> The results presented above demonstrate that by increasing the batch size, we can achieve a more substantial acceleration in terms of throughput compared to the baseline model.
>
> **Q4: Why Ailurus performs better than existing methods when applied to shallower layers?**
>
> **A4:** Thanks for your question! As we discussed in the general response, Ailurus reconstructs tokens using the most similar ones, and their similarity could be preserved even after several transformer blocks. In contrast, Expedite reconstructs tokens using the super-pixel features, which are less similar to the original tokens. The limited similarity makes it hard to preserve their relations when passing through transformer blocks (the more passed blocks, the less similarity is preserved) and causes more serious performance degradation compared with Ailurus. Thus, Ailurus performs much better than Expedite when applied to shallower layers as passing more blocks.
>
> **Q5: How does the proposed method recover the output of each transformer block?**
>
> **A5:** We recover the spatial resolution of the feature map through the unmerge operation. Specifically, we store the produced assignments and unfold merged tokens at output layers.

---

> > ### Comment · Area_Chair_XNw7 · 2023-08-19
> > **Has the rebuttal addressed your concerns?**
> >
> > Dear Reviewer PvMg,
> >
> > Could you please read the author rebuttal and acknowledge if your concerns have been addressed? The discussion period will end very soon on Monday, August 21. Thank you for your time in reviewing this submission!
> >
> > Best,
> >
> > AC

---

> > ### Comment · Reviewer_PvMg · 2023-08-21
> > **Feedback after the rebuttal**
> >
> > Thanks for the authors' careful feedback. I have read the rebuttal and some of the concerns are addressed, e.g., time consumption, output recovery, etc. However, some of my questions still exist, i.e. the novelty, the impact of $\alpha$, and the batch inference.
> >
> > 1. The difference between the TCFormer and the proposed method is not significant. It is better to compare the TCFormer and the Ailurus in different dense prediction tasks.
> >
> > 2. It is better to present the inference speed of Ailurus with batch 16 for a fair comparison.
> >
> > Above all, I will keep the original rating.

---

> > > ### Author Response · Authors · 2023-08-21
> > >
> > > We appreciate your thoughtful feedback and would like to address your remaining concerns as follows:
> > >
> > > **Comparison with TCFormer**
> > >
> > > 1. It's important to highlight that TCFormer relies on fine-tuning, making it inaccessible for well-trained models. On the other hand, our proposed model, Ailurus, can be integrated into well-trained models to instantly accelerate inference. For instance, we utilized the recent text-to-video generation model, ControlVideo\[1\], as a case study to exemplify Ailurus' versatility. We incorporated Ailurus into ControlVideo, which resulted in halving the number of tokens during inference. The accordingly generated results are provided in the supplementary PDF file. Specifically, Ailurus maintains a comparable quality to the original model while halving the time required. Its flexibility and parameter-free characteristics allow Ailurus to be applied in a broader range of scenarios where TCFormer cannot be utilized.
> > >
> > > 2. As mentioned in our general response, replacing the DPC algorithm with a naive strategy in the Ailurus framework still yields satisfactory performance. This demonstrates that the DPC algorithm isn't a prerequisite for Ailurus to function effectively. The key to Ailurus' robustness and superiority lies in its framework, which manages to preserve the relationships between the remaining and merged tokens with little distortion.
> > >
> > > 3. As TCFormer is designed for human-centric tasks, its application to dense prediction tasks presents numerous challenges. Specifically, TCFormer relies on specific architectures and training, limiting its usability for direct comparison with Ailurus in well-trained models. Additionally, the unique architecture of TCFormer makes it difficult to leverage the benefits of extensive pre-trained models within the ViT-family. Consequently, comparing these two methods becomes challenging and less meaningful due to their distinct application scenarios. Nevertheless, we are diligently working to adapt TCFormer for compatibility with dense prediction tasks and conduct a comparative analysis against Ailurus. As the discussion period is drawing to a close, this task is challenging, but we will invest our best efforts to address it. If time permits, we will promptly report our results.
> > >
> > > **Performance Comparison with Batch Size 16**
> > >
> > > Below we present the performance comparison using a batch size of 16. As you will see, Ailurus still demonstrates outstanding performance.
> > >
> > > | Method   | Num clusters | Batch size |          FPS           |       Throughput        | mIoU  |
> > > | :------- | :----------: | :--------: | :--------------------: | :---------------------: | :---: |
> > > | Baseline |     1024     |     16     |          32.2          |          35.6           | 48.48 |
> > > | Ailurus  |     324      |     16     | 61.2 ($\uparrow$ 90%) | 76.1 ($\uparrow$ 114%) | 46.45 |
> > > |          |     400      |     16     | 53.7 ($\uparrow$ 67%) | 67.4 ($\uparrow$ 89%)  | 47.12 |
> > >
> > > We are open to further questions and sincerely look forward to your response.
> > >
> > > \[1\]. Zhang, Yabo, et al. "ControlVideo: Training-free Controllable Text-to-Video Generation." arXiv preprint arXiv:2305.13077 (2023).

---

> > > ### Author Response · Authors · 2023-08-21
> > >
> > > **TCFormer for dense prediction tasks**
> > >
> > > We follow your suggestion to apply TCFormer to dense prediction tasks. Initially, we intend to integrate TCFormer into the mmseg framework for semantic segmentation. We carefully implement the TCFormer under the mmseg framework according to the officially released code. However, we find that TCFormer is an extremely computationally intensive backbone. Even with its lightest configuration, the FPS for inferring ADE20K images at a resolution of 512x512 under the Segmenter framework is only 2.03. In comparison, a typical ViT-Base model achieves an FPS of 32.2, while Ailurus can surpass 50 without sacrificing performance. Upon analyzing the TCFormer code, we discover that it heavily relies on sparse matrix multiplication. Although these operations may not have high FLOPs, they introduce significant latency. Therefore, it is nearly impossible and impractical to apply TCFormer to dense prediction tasks given its high computational complexity.
> > >
> > > Furthermore, we attempt to integrate the core TCFormer design, the CTM module, into ViT for acceleration. This would allow us to compare the performance of TCFormer with Ailurus using the same backbone and initialization. However, we encountered the same issue of excessive complexity for the CTM module. In a similar setup, the time required for the CTM module to execute is even higher than the inference time of the model itself. The details are presented below:
> > >
> > > | Method  | Total (ms) | blocks (ms) | extra (ms) |  mIoU |
> > > |:------| :---------:|:----------:|:--------:|:-----:|
> > > |TCFormer   | 33.8   |   8.6      | 25.2        | 0.14 |
> > > |Percents       | -          | 25.0%        |  75.0%  | - |
> > > |Ailurus     | 10.0      | 8.7        | 1.3        | 47.35 |
> > > |Percents       | -          | 87.0%        | 13.0%        | - |
> > >
> > > Despite these issues, we proceed to train the ViT model with the integrated CTM module in the Segmenter framework for comparison with Ailurus. We followed the instructions provided in the TCFormer paper to set up the CTM module and trained ViT-CTM with the same optimization parameters and initialization as Ailurus. However, we find that even with only the integration of CTM into ViT, it raises a significant negative impact on training and causes very poor performance. Thus, it is untrivial to integrate the CTM module to ViTs for acceleration.
> > >
> > > |Method   | Num clusters  |  FPS   | mIoU |
> > > |:------| :---------:|:----------:|:--------:|
> > > |Baseline   | 1024    |   32.2        | 49.60  |
> > > |ViT-CTM  | 410 (1024 * 0.4)     | 29.1 ($\downarrow$ 10%)             | 33.64 ($\downarrow$ 15.96)  |
> > > |Ailurus    | 400      |  53.7 ($\uparrow$ 67%)           | 49.04 ($\downarrow$ 0.56)  |
> > >
> > > In conclusion, TCFormer is specifically designed for human-centric tasks, and adapting it to dense prediction tasks poses significant challenges. Additionally, TCFormer exhibits high computational complexity, rendering it unsuitable for accelerating ViTs. In contrast, Ailurus can be seamlessly integrated into well-trained models, providing immediate acceleration or expediting training without the need for additional hyper-parameter adjustments. These two methods differ significantly in terms of motivation, implementation, and application. Ailurus stands out for its flexibility and lightweight nature, enabling its deployment in various tasks that TCFormer can't accomplish.
> > >
> > > We sincerely appreciate your diligent review of this paper, and we would be grateful to receive any response or feedback from you.

---

> > > ### Author Response · Authors · 2023-08-22
> > >
> > > Dear reviewer PvMg:
> > >
> > > In the past day, we made our best efforts to conduct a comparison between TCFormer and Ailurus for dense prediction tasks. The results indicate that TCFormer is not suitable for dense prediction tasks and is unable to surpass Ailurus. For detailed information, please refer to our response.
> > >
> > > We would greatly appreciate it if our response could effectively address any of your concerns, and we would be grateful for any feedback or response from you.

---

### Official Review · Reviewer_ZFoB · 2023-07-09

**Soundness:** 3 good
**Presentation:** 3 good
**Contribution:** 2 fair
**Rating:** 5
**Confidence:** 3

**Summary:**

This paper proposes a new approach to improving the efficiency of Vision Transformers (ViTs) for dense prediction tasks such as semantic segmentation. Recognizing that these tasks emphasize the contours or shapes of objects more than the textures within, the authors introduce a method to apply adaptive resolution to tokens in ViTs through density-based clustering. At the intermediate layer of the ViT, a spatial-aware density-based clustering algorithm is used to select anchors from the token sequence. Tokens near these anchors are merged to form low-resolution regions, while others are kept as high-resolution, thereby significantly reducing the number of tokens processed in subsequent layers. This strategy aims to accelerate processing without loss of critical information.

The proposed method showcased promising results on different semantic segmentation datasets compared to a couple of previous works, including a baseline method [13] which also conducts token clustering. The proposed method provides higher speed up with less performance drop on both well-trained and fine-tuning scenarios.

**Strengths:**

- The paper presents promising empirical results compared to a closely related baseline [13], especially in reducing fine-tuning training time.
- The paper is well-intuitive and easy to follow.
- The paper provides extensive analysis and ablation on parameters, cluster numbers, and clustering layers.
- The paper provides visual analysis to demonstrate that the clustering of tokens towards the segmentation task is reasonable.

**Weaknesses:**

- The manuscript suffers from a substantial number of typographical errors. Some are actually critical. E.g., the abusing usage of delta and sigma (Ln138-142).
- The paper can be improved by better structure and writing. E.g., The figures and Tables are all over the place instead of being roughly around the page when they are mentioned.
- While I recognize the promising performance of the paper, I am concerned that the paper might not contain enough novelty and theory contribution for publication in the underlying conference. The idea to decrease the number of tokens has been proposed in related works, as mentioned in the paper itself. The clustering idea is also proposed by [13].  The critical contribution of the paper is adapting a different existing density-based clustering method to the semantic segmentation task. In addition, I don't think the paper provides enough concrete analysis on why the proposed clustering method is better than the one in [13], despite empirical results.
- The paper refers to the text's segmentation and object detection, while only semantic segmentation results are presented in the main experiment section. For the sake of fairness, I would recommend removing the misleading reference to the object detection task or using it for future work.
- The visualization example doesn't contribute much to the paper as the idea of decreasing the token for tasks that do not need local fine-gran information has been proposed and demonstrated in related works. Plus, the figures are not quite self-evident. For E.g. the clustering validity can depend on ontology.

**Questions:**

- Does clustering location refers to layer number? If so, please declare.
- Any idea why the new clustering method provides more robust clustering than [13]?
- The ablation study seems to indicate that alpha makes little difference, any thoughts on that?

**Limitations:**

I don't think there are any outstanding concerns about any negative societal impacts in this work.

---

> ### Author Rebuttal · Authors · 2023-08-10
>
> **Q1: Typographical errors**
>
> **A1:** We are sorry for these typos and have corrected them in the revised manuscript. Many thanks for your patience.
>
> **Q2: The placements of figures and tables**
>
> **A2:** We sincerely appreciate your suggestion and will rearrange them to make it easier to refer to.
>
> **Q3: The motivation**
>
> **A3:** We have discussed our motivation in our main text (Line 33-42) and the general response. Please refer to the general response.
>
>
> **Q4: Why does Ailurus perform better than previous methods?**
>
> **A4.**  As illustrated in our experiments and **the general response**, Ailurus applies a more reasonable approach for feature reconstruction. This ensures the superior performance of Ailurus compared to Expedite, especially in challenging scenarios. Please refer to the general response for more details
>
>
> **Q5: Object detection results**
>
> **A5:** You may have missed the object detection and the instance segmentation experiments in our appendix, for which we apologize. We summarize the results here for your reference and have added them to the main text in the revised manuscript. Specifically, we integrate Ailurus into well-trained Mask-RCNN models for both object detection and instance segmentation and conduct experiments over various configurations and backbones. The results show that Ailurus still performs well in other dense prediction tasks and are presented below.
>
>
> Results for ViT-B:
>
> |    | Location   | Clusters   | FPS                  | Object Detection        | Instance Segmentation   |
> |:----------|:-----------|:-----------|:---------------------|:------------------------|:------------------------|
> | Baseline  | -          | -          | 4.4                  | 50.1                    | 44.0                    |
> |    | 3          | 1296       | 6.3 ($\uparrow$ 43%) | 48.3 ($\downarrow$ 1.8) | 42.4 ($\downarrow$ 1.6) |
> |    | 3          | 1764       | 5.9 ($\uparrow$ 34%) | 49.4 ($\downarrow$ 0.7) | 43.4 ($\downarrow$ 0.6) |
> |    | 6          | 1444       | 5.3 ($\uparrow$ 20%) | 49.8 ($\downarrow$ 0.3) | 43.9 ($\downarrow$ 0.1) |
> |    | 7          | 1444       | 5.1 ($\uparrow$ 16%) | 49.9 ($\downarrow$ 0.2) | 43.9 ($\downarrow$ 0.1) |
>
> Results for ViT-L:
>
> |    | Location   | Clusters   | FPS                  | Object Detection        | Instance Segmentation   |
> |:----------|:-----------|:-----------|:---------------------|:------------------------|:------------------------|
> | Baseline  | -          | -          | 1.9                  | 54.7                    | 47.8                    |
> |    | 7          | 1600       | 2.9 ($\uparrow$ 53%) | 53.0 ($\downarrow$ 1.7) | 46.5 ($\downarrow$ 1.3) |
> |    | 9          | 1764       | 2.6 ($\uparrow$ 37%) | 54.0 ($\downarrow$ 0.7) | 47.2 ($\downarrow$ 0.6) |
> |    | 11         | 1764       | 2.5 ($\uparrow$ 32%) | 54.3 ($\downarrow$ 0.4) | 47.4 ($\downarrow$ 0.4) |
> |    | 15         | 1600       | 2.3 ($\uparrow$ 21%) | 54.5 ($\downarrow$ 0.2) | 47.6 ($\downarrow$ 0.2) |
>
> As presented above, Ailurus still performs well in object detection and instance segmentation tasks, which demonstrates its generalization ability.
>
> **Q6: The visualizations**
>
> **A6:** We respectfully disagree with this comment. As mentioned in the general response, Ailurus focuses on dense prediction tasks, which receive less attention in previous related works. Dense prediction tasks quite rely on local fine-grain information, especially for semantic segmentation tasks which are required to produce pixel-level prediction. Thus, this paper is not so-called "decreasing the token for tasks that do not need local fine-grain information". On the contrary, this paper focuses on how to preserve local fine-grain information with as less as possible tokens. To this end, we propose the adaptive resolution strategy, and the most critical component is the resolution assignments. This visualization shows that Ailurus manages to preserve local fine-grain information with only 1/4 patches, which clearly demonstrates the effectiveness of the generated assignments. The visualizations strongly support our motivation and explain why Ailurus could accelerate vision transformers with negligible performance loss. As commented by Reviewer4 2MFv, "This visualization in Fig 2. is one of the most important ones in this paper".
>
>
> Besides, the provided visualizations contain different scenarios, and some of them are quite complicated such as 4-th, 6-th, and 8-th. These diverse and complicated scenarios lend credence to the visualization results.
>
> **Q7: Clustering location**
>
> **A7:** Yes, the clustering location indicates the layer index where we perform clustering. We have declared it in our revised manuscript. Thanks for your suggestion!
>
> **Q8: Why does Ailurus provide more robust clustering than Expedite?**
>
> **A8:** Thanks for your question! As we discussed in **the general response**, Ailurus reconstructs tokens depending on similar tokens, and their similarity could be preserved even after several transformer blocks. In contrast, Expedite reconstructs tokens using the super-pixel features, which are less similar to the original tokens. The limited similarity makes it hard to preserve their relations when passing through transformer blocks and causes more serious performance degradation compared with Ailurus.
>
> **Q9: The impact of $\alpha$.**
>
> **A9:** Please note that, as described in the context, the variable $\alpha$ represents the penalty strength applied to tokens within the neighborhood. A smaller value of $\alpha$ indicates a larger penalty strength, indicating a preference for prioritizing closer tokens. Since the tokens are already constrained within a neighborhood, they inherently exhibit relatively high similarity, resulting in relatively low variance for different values of $\alpha$.

---

> > ### Comment · Area_Chair_XNw7 · 2023-08-19
> > **Has the rebuttal addressed your concerns?**
> >
> > Dear Reviewer ZFoB,
> >
> > Could you please read the author rebuttal and acknowledge if your concerns have been addressed? The discussion period will end very soon on Monday, August 21. Thank you for your time in reviewing this submission!
> >
> > Best,
> >
> > AC

---

### Official Review · Reviewer_gX5G · 2023-07-10

**Soundness:** 3 good
**Presentation:** 3 good
**Contribution:** 3 good
**Rating:** 6
**Confidence:** 4

**Summary:**

This paper studies the problem of accelerating vision transformers for dense prediction without fine-tuning and proposes a spatial-aware density-based clustering algorithm to select anchors from the token sequence. The proposed method is evaluated across three different
datasets and demonstrates promising performance.

**Strengths:**

+ The idea is reasonable and interesting, and the performance is good.
+ The abundant ablation studies are provided to verify the effectiveness of the proposed method.

**Weaknesses:**

- As said in the title, the proposed method is for dense prediction, of course, segmentation is typical, But it is better to include another dense prediction task for further verifying the effectiveness of the proposed method, for example, instance segmentation, depth estimation.
- It is better to calculate the reconstruction error between the reconstructed feature map and the original feature map. Meanwhile, make a comparison to the other SOTA method, e.g. Expedite.

**Questions:**

Please refer to Weaknesses

---

> ### Author Rebuttal · Authors · 2023-08-10
>
> **Q1: Other dense prediction tasks like instance segmentation and depth estimation**
>
> **A1:** Thanks for your suggestion! We have presented the performance on object detection and instance segmentation in the appendix and summarize the results here for your reference. Specifically, we integrate Ailurus into well-trained Mask-RCNN models for both object detection and instance segmentation and conduct experiments over various configurations and backbones. The results are presented below.
>
> Results for ViT-B:
>
> |    | Location   | Clusters   | FPS                  | Object Detection        | Instance Segmentation   |
> |:----------|:-----------|:-----------|:---------------------|:------------------------|:------------------------|
> | Baseline  | -          | -          | 4.4                  | 50.1                    | 44.0                    |
> |    | 3          | 1296       | 6.3 ($\uparrow$ 43%) | 48.3 ($\downarrow$ 1.8) | 42.4 ($\downarrow$ 1.6) |
> |    | 3          | 1764       | 5.9 ($\uparrow$ 34%) | 49.4 ($\downarrow$ 0.7) | 43.4 ($\downarrow$ 0.6) |
> |    | 6          | 1444       | 5.3 ($\uparrow$ 20%) | 49.8 ($\downarrow$ 0.3) | 43.9 ($\downarrow$ 0.1) |
> |    | 7          | 1444       | 5.1 ($\uparrow$ 16%) | 49.9 ($\downarrow$ 0.2) | 43.9 ($\downarrow$ 0.1) |
>
> Results for ViT-L:
>
> |    | Location   | Clusters   | FPS                  | Object Detection        | Instance Segmentation   |
> |:----------|:-----------|:-----------|:---------------------|:------------------------|:------------------------|
> | Baseline  | -          | -          | 1.9                  | 54.7                    | 47.8                    |
> |    | 7          | 1600       | 2.9 ($\uparrow$ 53%) | 53.0 ($\downarrow$ 1.7) | 46.5 ($\downarrow$ 1.3) |
> |    | 9          | 1764       | 2.6 ($\uparrow$ 37%) | 54.0 ($\downarrow$ 0.7) | 47.2 ($\downarrow$ 0.6) |
> |    | 11         | 1764       | 2.5 ($\uparrow$ 32%) | 54.3 ($\downarrow$ 0.4) | 47.4 ($\downarrow$ 0.4) |
> |    | 15         | 1600       | 2.3 ($\uparrow$ 21%) | 54.5 ($\downarrow$ 0.2) | 47.6 ($\downarrow$ 0.2) |
>
> As presented above, Ailurus still performs well in object detection and instance segmentation tasks, which demonstrates its generalization ability.
>
> **Q2: The reconstruction error and comparison to Expedite**
>
> **A2:** We sincerely appreciate your suggestion.
> For a comprehensive comparison, we run various configurations for Ailurus and Expedite on ViT-B and ViT-L, respectively, and calculate the cosine similarity between the reconstructed features and the original ones, where, 18 $\times$ 18, 20 $\times$ 20, 22 $\times$ 22, and 24 $\times$ 24 are different cluster numbers. The results are presented below. Ailurus consistently outperforms Expedite in terms of reconstruction errors across various settings, which partially explains why Ailurus performs better than Expedite. We also post more analysis for the reconstruction comparison between Ailurus and Expedite in the general response. Please refer to it for more details.
>
> Results for ViT-B
>
> | layer index | Method  | 18 $\times$ 18 |            | 20 $\times$ 20 |            | 22 $\times$ 22 |            | 24 $\times$ 24 |            |
> | :---------- | :------ | -------------: | ---------: | -------------: | ---------: | -------------: | ---------: | -------------: | ---------: |
> |             |         |           mIoU | Similarity |           mIoU | Similarity |           mIoU | Similarity |           mIoU | Similarity |
> | 2           | Ailurus |          46.45 |     0.8971 |          47.12 |     0.9291 |          47.70 |     0.9436 |          48.26 |     0.9602 |
> |             | Expedit |           4.32 |     0.2882 |           5.21 |     0.3342 |           6.19 |     0.3652 |           7.57 |     0.4065 |
> | 4           | Ailurus |          47.69 |     0.9339 |          48.27 |     0.9503 |          48.66 |     0.9636 |          48.88 |     0.9746 |
> |             | Expedit |          12.66 |     0.4714 |          15.53 |     0.5218 |          18.03 |     0.5594 |          21.57 |     0.6017 |
> | 6           | Ailurus |          48.19 |     0.9527 |          48.83 |     0.9642 |          48.90 |     0.9736 |          48.99 |     0.9817 |
> |             | Expedit |          44.07 |     0.8782 |          45.17 |     0.8972 |          45.44 |     0.9068 |          46.14 |     0.9161 |
>
> Results for ViT-L
>
> | layer index   | Method   |   20 $\times$ 20 |        |   22 $\times$ 22 |        |   24 $\times$ 24 |        |   26 $\times$ 26 |        |
> |:--------------|:---------|-----------------:|-------:|-----------------:|-------:|-----------------:|-------:|-----------------:|-------:|
> |               |          |  mIoU            |Similarity|   mIoU           |Similarity|   mIoU           |Similarity|    mIoU           |Similarity|
> | 4             | Ailurus  |            47.36 | 0.8401 |            48.84 | 0.8699 |            50.05 | 0.8941 |            50.56 | 0.9143 |
> |               | Expedit  |            42.88 | 0.7706 |            44.26 | 0.7926 |            47.19 | 0.8308 |            47.74 | 0.8504 |
> | 6             | Ailurus  |            48.80 | 0.8724 |            49.47 | 0.8967 |            50.45 | 0.9164 |            50.98 | 0.9326 |
> |               | Expedit  |            45.32 | 0.8162 |            46.05 | 0.8304 |            47.80 | 0.8626 |            48.37 | 0.8789 |
> | 8             | Ailurus  |            49.97 | 0.9039 |            50.39 | 0.9213 |            51.06 | 0.9356 |            51.37 | 0.948  |
> |               | Expedit  |            49.73 | 0.8934 |            50.03 | 0.9048 |            50.77 | 0.9219 |            51.06 | 0.9324 |

---

> > ### Comment · Area_Chair_XNw7 · 2023-08-19
> > **Has the rebuttal addressed your concerns?**
> >
> > Dear Reviewer gX5G,
> >
> > Could you please read the author rebuttal and acknowledge if your concerns have been addressed? The discussion period will end very soon on Monday, August 21. Thank you for your time in reviewing this submission!
> >
> > Best,
> >
> > AC

---

> > > ### Comment · Reviewer_ZFoB · 2023-08-21
> > >
> > > I appreciate the rebuttal from the author. I read it th
> > > The answers addressed some of my concerns.
> > > I would recommend adding more discussion on `why` the proposed method is better than [3], in additional to experimental results.
> > > I increased my rating accordingly.

---

> > > > ### Author Response · Authors · 2023-08-21
> > > >
> > > > We sincerely appreciate your response. Our general response analysis provides a detailed analysis of the advantage of Ailurus, and the results indicate that Ailurus only aggregates tokens with high intra-similarity and thus could preserve their relations going through transformer blocks. At the same time, Expedite [13] suffers from severe relation distortion during forward. As a result, Ailurus could recover the feature map at the output layer with little distortion and performs much better than Expedite [13].
> > > >
> > > > We also recognize that this is a critical contribution of Ailurus and will add this discussion to our later revisions. Sincerely appreciate for your suggestion!

---

### Author Rebuttal · Authors · 2023-08-10

We sincerely appreciate all reviewers for their efforts, and first give general responses for some common issues.

**Why does Ailurus performs better than Expedite?**

As presented in the original paper, Ailurus significantly outperforms Expedite, especially in cases **when performing token reduction at shallow layers**, which is more challenging as it suffers more serious information loss. We explain the reasons with detailed experiments to confirm the claim.

i) **Comparing the reconstruction errors of Ailurus and Expedite**. As posted in the response to Reviewer1 gx5G, Ailurus enjoys much lower reconstruction errors compared with Expedite across various settings, which partially explains why Ailurus performs better than Expedite.

ii) We further observe that **Ailurus exhibits more pronounced advantages in shallow clustering while Expedite's performance deteriorates a lot**. Since both Ailurus and Expedite perform clustering only once and reuse the clustering results at the output layer, it is evidently easier to perform clustering in deeper layers compared to shallow layers.  To investigate the reasons behind the performance gap in shallow clustering, we take a representative scenario (ViT-B, layer index=2, num cluster=20*20), and calculate the feature reconstruction quality in subsequent layers. The results are presented below. The results indicate that Expedite suffers from serious reconstruction distortion during the forward pass while Ailurus preserves the reconstruction quality across layers.

| Method  | Layer-2 | Layer-3 | Layer-4 | Layer-5 | Layer-6 | Layer-7 | Layer-8 | Layer-9 | Layer-10 | Layer-11 |
| :------ | ------: | ------: | ------: | ------: | ------: | ------: | ------: | ------: | -------: | -------: |
| Ailurus |  0.9210 |  0.8959 |  0.8863 |  0.8906 |  0.8972 |  0.9092 |  0.9238 |  0.9312 |   0.9361 |   0.9291 |
| Expedit |  0.7334 |  0.6228 |  0.5524 |  0.4824 |  0.4293 |  0.3694 |  0.3778 |  0.4219 |   0.4433 |   0.3342 |

iii) We analyze **the advantages of Ailurus over Expedite in terms of intrinsic mechanism**: both methods target at reducing the number of tokens while preserving the relationships between the retained tokens and the reduced ones, and then utilizing these relationships for recovery at the output layer. The quality of the recovery process determines the performance, and **the recovery quality depends on whether the preserved relationships can be maintained after passing through several Transformer blocks**. Expedite and Ailurus employ different methods to preserve these relationships.

**Compared with Expedite, the similarity between the aggregated tokens and the original tokens in Ailurus are higher.** For Expedite makes use of K-means clustering, while Ailurus directly aggregates low information density regions within the image (i.e., regions with high redundancy). To validate this empirically, we conducted a study on the similarity between the original tokens and the super-pixel features in Expedite, as well as the similarity between the aggregated tokens and the original tokens in Ailurus during the inference process. **The results showed that the average similarity in Expedite was 0.6728, while in Ailurus, it was 0.9210, which aligns with our expectations**.

We also investigate the probability of preserving the relationships at different levels of similarity at the output layer. The results are as follows. The results demonstrate that for tokens with a similarity between 0.9 and 1.0, there is a probability of over 99% that they will maintain this similarity at the output layer. However, tokens in other similarity ranges suffer from significant similarity distortion. This explains why Ailurus outperforms Expedite.

|   Intervals | 0.0-0.1 | 0.1-0.2 | 0.2-0.3 | 0.3-0.4 | 0.4-0.5 | 0.5-0.6 | 0.6-0.7 | 0.7-0.8 | 0.8-0.9 | 0.9-1.0 |
| ----------: | ------: | ------: | ------: | ------: | ------: | ------: | ------: | ------: | ------: | ------: |
| Probability |  0.0137 |  0.1246 |  0.2179 |  0.1439 |  0.0629 |  0.0347 |  0.0255 |  0.0193 |  0.0212 |  0.9903 |

**The motivation of Ailurus.**
1.  While there have been studies on accelerating Vision Transformers (ViTs) through token reduction, most of them have focused on classification tasks. This is because reducing the number of tokens affects the spatial resolution of feature maps, making them unsuitable for dense prediction tasks. In contrast, our method focuses on accelerating dense tasks, which is crucial for the efficient deployment of ViTs.

2.  Expedite attempts to accelerate dense prediction tasks by **replacing original features with fewer superpixel features**. However, as presented in our main text (Line 30-32), Expedite's acceleration is limited because it can only be deployed in relatively deep ViT layers.
We provide a concrete analysis of this issue in the general response 1, and our findings suggest that this limitation is due to the similarity between superpixel features and original features becoming lower and lower after several transformer blocks.

3.   The limitation of Expedite inspires us to consider **operating tokens in the original feature space**. Based on the **heterogeneous information distribution** in the image, we propose to apply adaptive resolution for features of different regions. Specifically, we use fewer tokens for regions with lower information density, while more tokens for regions containing local fine-grain information. Thus we can minimize the number of tokens while preserving performance in downstream dense prediction tasks. To achieve this goal, the core is to measure the information density. Therefore, we have chosen an algorithm based on **local spatial density**, i.e., DPC, to allocate the resolution in different regions of the image.

---

### Decision · Program_Chairs · 2023-09-21

**Decision:**

Accept (poster)

**Comment:**

This paper received mixed reviews.

The reasons to accept the paper:

- Target an important problem: speeding up dense prediction of ViT models for both training and inference.
- Good performance on standard benchmarks across different tasks: object detection, instance segmentation, and semantic segmentation.
- Abundant ablation studies with both quantitative and visual analysis.

The reason to reject the paper is because the idea is not novel enough. Specifically, the reviewers ZFoB, PvMg, and viNG pointed out that the DPC-based token clustering module has been proposed in the previous method[29] for transformer-based dense prediction tasks. The reviewer ZFoB is concerned the clustering idea is also proposed by [13] and not enough analysis has been shown why the proposed clustering method is better than [13].

In the rebuttal, the authors provided more empirical results showing

- Better performance and analysis of the proposed apporach than Expedite [13]. The proposed approach exhibits more advantages in shallow clustering while Expedite's performance deteriorates a lot.
- TCFormer [29] is specifically designed for human-centric tasks, and adapting it to dense prediction tasks poses significant challenges.

After the rebuttal, reviewers PvMg and viNG recommend Borderline Reject (viNG did not participate in the discussions with the authors).

The AC acknowledges that the DPC-based token clustering module has been studied in [29]. But the rebuttal shows that adapting it to dense prediction task is not trivial, making the proposed approach valuable. Overall, the AC finds the positive side of reviews outweights the negative side and thus recommends an acceptance as a poster.